

# The 3d twisted index and wall-crossing

Mathew Bullimore[1], Andrea E. V. Ferrari[2] and Heeyeon Kim[2]

**1** Department of Mathematics, Durham University
Lower Mountjoy, Stockton Road, Durham, DH1 3LE, UK
**2** Mathematical Institute, University of Oxford, Andrew Wiles Building,
Radcliffe Observatory Quarter, Woodstock Road, Oxford, OX2 6GG, UK

## Abstract

We study the twisted index of 3d $\mathcal{N} = 2$ supersymmetric gauge theories on $S^1 \times \Sigma$ in the presence of a real FI parameter deformation. This parameter induces a 1d FI parameter for the effective supersymmetric quantum mechanics on $S^1$. Using supersymmetric localisation, the twisted index can be expressed as a contour integral. We show that the contour prescription is modified in the presence of the 1d FI parameter, leading to wall-crossing phenomena for the twisted index. In particular, we derive a general wall-crossing formula for abelian gauge theories. We also examine the origin of wall-crossing as change of stability condition in the algebro-geometric interpretation of the twisted index. These ideas are illustrated for abelian theories with $\mathcal{N} = 4$ supersymmetry and in a non-abelian example that reproduces wall-crossing phenomena associated to moduli spaces of stable pairs.

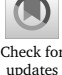

# 1  Introduction

This paper concerns the twisted index of 3d $\mathcal{N}=2$ gauge theories on $S^1 \times \Sigma$, where $\Sigma$ is a closed Riemann surface of genus $g$. The twisted index was first studied in the context of the Bethe/gauge correspondence [1], while contour integral formulae for the twisted index were derived using supersymmetric localisation for $g=0$ in [2] and extended to $g>0$ in [3,4]. This has subsequently found beautiful applications to exact microstate counting for supersymmetric black holes in AdS$_4$ [5–7].

It is natural to regard a twisted 3d $\mathcal{N}=2$ theory on $S^1 \times \Sigma$ as an $\mathcal{N}=(0,2)$ supersymmetric quantum mechanics on $S^1$. The twisted index is then identified with the Witten index [8] of the supersymmetric quantum mechanics. Provided the spectrum is gapped, the twisted index can be expressed as

$$\mathcal{I} = \sum_{d \in \pi_1(G)} q^d \, \mathrm{Tr}_{\mathcal{H}_d}(-1)^F y^{G_F} \,, \tag{1}$$

for any compact connected gauge group $G$. The summation is over the topological class $d \in \pi_1(G)$ of the principal $G$-bundle on $\Sigma$, weighted by a fugacity $q$ for the topological global symmetry. The trace is over supersymmetric ground states $\mathcal{H}_d$ in each topological sector, graded by any additional flavour symmetry $G_F$ with fugacity $y$.

Gauged $\mathcal{N}=(0,2)$ quantum mechanics exhibit wall-crossing phenomena in the space of 1d FI parameters, $\tau$. This is an exact parameter and so the Witten index is independent of $\tau$ provided the spectrum remains gapped. However, it may jump across real codimension-1

loci where a non-compact Coulomb branch opens up and the space of supersymmetric ground states $\mathcal{H}_d$ changes discontinuously. This wall-crossing phenomenon has found important applications. For example, D-particles in Type II superstring theory enjoy an effective description in terms of quiver quantum mechanics, and the BPS spectrum of boundstates jumps according to the quantum mechanical wall-crossing [9]. In [10] the quantum-mechanical wall-crossing was systematically studied from the point of view of supersymmetric localisation.

The purpose of this paper is to evaluate the twisted index on $S^1 \times \Sigma$ in the presence of a real 1d FI parameter $\tau$ and explore wall-crossing in this context based on different localisation schemes. The 1d FI parameter can in principle be induced by a real 3d FI parameter, but one can also treat it as an independent parameter. The dependence on this parameter is new and should be distinguished from the known dependence of the twisted index on the complexified 3d FI parameter $\zeta$, which does not display wall-crossing phenomena [11]. One motivation to perform this study is to obtain an effective quantum-mechanical description that is valid for each topological sector $d$. This effective description should put the evaluation of the twisted index in the same realm of wall-crossing phenomena in quasi-map theory [12, 13].

The first goal is to explain how the contour integral formulae derived in [2–4] are modified, if at all, in the presence of $\tau$. For illustration, we will focus here on $G = U(1)$. The twisted index is expressed using a Jeffrey-Kirwan residue prescription,

$$\mathcal{I} = \sum_{d \in \mathbb{Z}} q^d \sum_{x_*} \underset{x=x_*}{\text{JK-Res}}(Q_{x_*}, \eta) \frac{dx}{x} g_d(x, y), \tag{2}$$

where the integrand $g_d(x, y)$ is a rational function that includes contributions from 1-loop determinants and a gaugino zero-mode integral. The summation runs over $d \in \mathbb{Z}$ and poles $x_*$ of the integrand. The Jeffrey-Kirwan residue depends on a charge $Q_*$ associated to each pole and an auxiliary parameter $\eta \neq 0$.

Following arguments akin to [10], we will show that in the presence of the 1d FI parameter $\tau \in \mathbb{R}$ the charges $Q_+, Q_-$ assigned to the poles at the boundary $x \to 0, \infty$ are

$$Q_\pm = \begin{cases} \mp k_\pm^{\text{eff}} & \text{if} \quad k_\pm^{\text{eff}} \neq 0 \\ d - \widetilde{\tau} & \text{if} \quad k_\pm^{\text{eff}} = 0 \end{cases}, \tag{3}$$

where $k_\pm^{\text{eff}}$ denote effective supersymmetric Chern-Simons levels and $\widetilde{\tau}$ is related $\tau$ by a constant positive normalisation. This differs from the prescription of [2–4] when $k_\pm^{\text{eff}} = 0$ and leads to the wall-crossing formula

$$\Delta \mathcal{I} = q^{d_*} \left[ \delta_{k_+^{\text{eff}}, 0} \underset{x=0}{\text{Res}} + \delta_{k_-^{\text{eff}}, 0} \underset{x=\infty}{\text{Res}} \right] \frac{dx}{x} g_{d_*}(x, y), \tag{4}$$

when $\widetilde{\tau}$ crosses the integer $d_* \in \mathbb{Z}$ from below.

An important consequence is that all charges $Q_*$ are non-vanishing provided $\widetilde{\tau} \notin \mathbb{Z}$. This means the twisted index is independent of the auxiliary parameter $\eta$ for each individual flux, without needing to sum over $d \in \mathbb{Z}$ as in [2–4]. This feature is necessary if there is to be an effective supersymmetric quantum mechanics for each $d \in \mathbb{Z}$. The original JK residue prescription of [2–4] is recovered in the limits $\widetilde{\tau} \to +\infty$ with fixed $\eta > 0$ or $\widetilde{\tau} \to -\infty$ with fixed $\eta < 0$, with the equivalence of these two limits amounts to the relation $\sum_{d \in \mathbb{Z}} q^d = 0$ for $q \neq 0$.

This opens up the possibility of an alternative localisation scheme leading to an effective supersymmetric quantum mechanics for each individual $d \in \pi_1(G)$. This is the approach to the twisted index taken in our previous papers [14, 15]. For illustration, we continue with $G = U(1)$. Provided $\widetilde{\tau} \notin \mathbb{Z}$, the path integral localises to two types of configurations. The first are solutions of the vortex equations on $\Sigma$,

$$*F_A + e^2(\mu(\phi) - \tau) = 0, \qquad \bar{\partial}_A \phi = 0, \tag{5}$$

where $\phi$ is non-vanishing and the gauge group is broken to a discrete subgroup. Here $\phi$ denotes the scalar components of chiral multiplets and $\mu(\phi)$ the moment map for the gauge action. The second are topological solutions where $\phi = 0$ and the gauge group is unbroken.

We denote the moduli space of solutions with fixed $d \in \mathbb{Z}$ by $\mathfrak{M}_{\tau,d}$. Algebraically, this parametrises a holomorphic line bundle $E$ of degree $d$ together with a holomorphic section $\phi$ of an associated holomorphic vector bundle, subject to a stability condition depending on $\tau$. For example, existence of the solutions to the vortex equations (5) maps to '$\tau$-stability' for the pair $(E, \phi)$. This correspondence has been extensively studied in the mathematical literature [16–19].

In line with the existence of an effective supersymmetric quantum mechanics in each sector $d \in \mathbb{Z}$, the twisted index can be expressed as a generating function

$$\mathcal{I} = \sum_{d \in \mathbb{Z}} q^d \int \hat{A}(\mathfrak{M}_{\tau,d}) \operatorname{Ch}(\mathcal{E}_{\tau,d}), \tag{6}$$

where $\mathcal{E}_{\tau,d}$ is in general a complex of coherent sheaves encoding contributions from Fermi multiplet zero modes and Chern-Simons terms. The moduli space $\mathfrak{M}_{\tau,d}$ should really be understood as an algebraic stack and equation (6) is an integral of virtual characteristic classes against the virtual fundamental class.

Sticking with $G = U(1)$, the moduli space $\mathfrak{M}_{\tau,d}$ can jump accross the wall $\tilde{\tau} = d$. From an algebraic perspective, this is due to a change of stability condition. If a non-compact Coulomb branch opens up, the twisted index can undergo wall-crossing. We show that this happens when $k_{\pm}^{\text{eff}} = 0$, in agreement with the modified JK residue prescription (3). For abelian theories with $\mathcal{N} = 4$ supersymmetry, which admit only vortex saddle points in our localisation scheme, we demonstrate precise agreement between the geometric picture (6) and the modified JK residue prescription in equation (2). The extension to more general $\mathcal{N} = 2$ theories with topological saddle points is discussed in [20].

We also study the generalisation of these ideas for non-abelian $G$, although less systematically. We consider a class of theories with $G = U(N)$, in which $\mathfrak{M}_{\tau}$ coincides with the moduli space of rank $N$ stable pairs [17]. This is known to have an intricate chamber structure in the parameter space $\tau \in \mathbb{R}$. For $N = 2$, the moduli space has been constructed explicitly in [18] and further studied in [19,21]. We show that the twisted index recovers the Hirzebruch genus of the moduli space of stable pairs [22] and is consistent with wall-crossing. We also comment on the generalisation to $N > 2$.

The paper is organised as follows. In section 2, we summarise the Lagrangians used in supersymmetric localisation and discuss how the twisted index depends on the parameters appearing in them. In section 3, we revisit the Coulomb branch localisation scheme of [5–7] in the presence of the 1d FI parameter and derive the modified JK residue prescription and abelian wall-crossing formula. In section 4, we consider an alternative localisation scheme leading to the algebro-geometric interpretation of the twisted index and demonstrate compatibility with wall-crossing. In section 5, we consider the example of abelian $\mathcal{N} = 4$ gauge theories, while in section 6, we explore a non-abelian example with a connection to the moduli spaces of stable pairs.

## 2 The Twisted Index

We consider a 3d $\mathcal{N} = 2$ gauge theory on $S^1 \times \Sigma$ with twist along an closed orientable Riemann surface $\Sigma$ of genus $g$ using an unbroken R-symmetry. This setup preserves two supercharges $Q, \bar{Q}$ generating a supersymmetric quantum mechanics on $S^1$. The supermultiplets are of the

type obtained by dimensional reduction of 2d $\mathcal{N} = (0,2)$ and we therefore refer to this as $\mathcal{N} = (0,2)$ quantum mechanics.

In this section, we review how 3d $\mathcal{N} = 2$ supermultiplets decompose under 1d $\mathcal{N} = (0,2)$ supersymmetry and the Lagrangians used in supersymmetric localisation. We also discuss how the twisted index depends on parameters appearing in these Lagrangians, including the 1d FI parameter $\tau$ that plays an important role in this paper. We generally follow the notation of reference [4].

## 2.1 Preliminaries

We consider a theory with compact connected gauge group $G$. Principal $G$-bundles on the Riemann surface $\Sigma$ are classified topologically by the fundamental group $\pi_1(G)$. The UV topological symmetry is then

$$G_T = \text{Hom}(\pi_1(G), U(1)), \tag{7}$$

which is also the centre of the Langlands dual group, $Z(^L G)$. Here we are assuming that $dF = 0$ identically, excluding theories with monopole operators in the superpotential. Given $d \in \pi_1(G)$, we denote the corresponding homomorphism $\text{Hom}(G_T, U(1))$ by $q \mapsto q^d$ for any element $q \in G_T$. Our canonical example will be $G = U(N)$, in which case $\pi_1(G) \cong \mathbb{Z}$, $G_T \cong U(1)$ and the notation $q^d$ is obvious.

## 2.2 Standard Lagrangians

Let us first consider a 3d $\mathcal{N} = 2$ vectormultiplet $(\sigma, A_\mu, \lambda_\alpha, \bar{\lambda}_\alpha, D)$ for the gauge group $G$. After twisting on $\Sigma$, this decomposes into two 1d $\mathcal{N} = (0,2)$ supermultiplets:

- A vector multiplet $(\sigma + iA_0, \lambda, \bar{\lambda}, D_{1d})$ for the group $\text{Aut}(P)$ of smooth gauge transformations of a principal $G$-bundle $P$ on $\Sigma$. It is important in the following that the 1d auxiliary field is $D_{1d} = D - 2F_{1\bar{1}}$.

- A chiral multiplet $(A_{\bar{1}}, \bar{\Lambda}_{\bar{1}})$ valued in $\Omega^{0,1}(\text{Ad} P)$ where $\text{Ad} P = P \times_G \mathfrak{g}$ is the associated vector bundle in the adjoint representation. More invariantly, the chiral multiplet parametrises the complex structure $\bar{\partial}_A$ on $\text{Ad} P$ induced by the gauge connection.

The supersymmetric Yang-Mills Lagrangian for the vectormultiplet is

$$\begin{aligned} L_{\text{YM}} = \text{tr}\Big[ &\frac{1}{2}F_{01}F_{0\bar{1}} + \frac{1}{2}(-2iF_{1\bar{1}})^2 + \frac{1}{2}D^2 + \frac{1}{2}|D_\mu \sigma|^2 - i\bar{\lambda}D_0\lambda - i\bar{\Lambda}_{\bar{1}}D_0\Lambda_1 \\ &+ 2i\bar{\Lambda}_{\bar{1}}D_1\lambda - 2i\Lambda_1 D_{\bar{1}}\bar{\lambda} - i\bar{\Lambda}_{\bar{1}}[\sigma, \Lambda_1] + i\bar{\lambda}[\sigma, \lambda]\Big] \end{aligned} \tag{8}$$

and coincides with the sum of the standard vectormultiplet and chiral multiplet Lagrangians for the 1d $\mathcal{N} = (0,2)$ supermultiplets above.

We can introduce a supersymmetric Chern-Simons term with level $k \in H^4(BG, \mathbb{Z})$. The Lagrangian is written schematically as

$$L_{\text{CS}} = \frac{k}{4\pi}\text{Tr}\Big[ i\epsilon^{\mu\nu\rho}\Big(A_\mu \partial_\nu A_\rho - \frac{2i}{3}A_\mu A_\nu A_\rho\Big) - 2D\sigma + 2i\bar{\lambda}\lambda + 2i\bar{\Lambda}_{\bar{1}}\Lambda_1 \Big], \tag{9}$$

where Tr is shorthand for a positive-definite bilinear form on $\mathfrak{g}$. For example, for $G = U(N)$ with $N > 1$ there are two independent levels associated to the simple and abelian summands in $\mathfrak{g} = \mathfrak{u}(N)$. We can also introduce a 3d FI parameter $\zeta$ valued in the Lie algebra of the topological symmetry $G_T$. The Lagrangian can be written as

$$L_{\text{FI}} = -\frac{i}{2\pi}\zeta \cdot D, \tag{10}$$

where we use that $\mathfrak{g}_T = Z(\mathfrak{g}^*) \subset \mathfrak{g}^*$ and the natural pairing between $\mathfrak{g}$ and $\mathfrak{g}^*$. As discussed further below, this is complexified by a background Wilson line for the topological symmetry along $S^1$.

We now consider a 3d $\mathcal{N} = 2$ chiral multiplet $\Phi = (\phi, \psi_\alpha, F)$ transforming in a faithful unitary representation $R$ of $G$ and R-charge $r$. After the topological twist this decomposes into two 1d $\mathcal{N} = (0,2)$ supermultiplets:

- A chiral multiplet $(\phi, \psi)$ valued in $\Omega^{0,0}(P_\Phi)$.

- A Fermi multiplet $(\eta, F)$ valued in $\Omega^{0,1}(P_\Phi)$.

Here we write

$$P_\Phi := K_\Sigma^{r/2} \otimes (P \times_G R), \tag{11}$$

for the associated vector bundle in the representation $R$, twisted by a power of the canonical bundle $K_\Sigma$, if necessary choosing a spin structure on $\Sigma$. The chiral multiplet Lagrangian is then

$$
\begin{aligned}
L_\Phi = \operatorname{tr}\big[ & \bar{\phi}(-D_0^2 - 4D_1 D_{\bar{1}} + \sigma^2 + iD - 2iF_{1\bar{1}})\phi - \bar{F}F \\
& -\frac{i}{2}\bar{\psi}(D_0 + \sigma)\psi - 2i\bar{\eta}(D_0 - \sigma)\eta + 2i\bar{\psi}D_1\eta - 2i\bar{\eta}D_{\bar{1}}\psi \\
& -i\bar{\psi}\bar{\lambda}\phi + i\bar{\phi}\lambda\psi - 2i\bar{\phi}\Lambda_1\eta + 2i\bar{\eta}\bar{\Lambda}_{\bar{1}}\phi \big]
\end{aligned}
\tag{12}
$$

and coincides with the standard Lagrangians for the above 1d $\mathcal{N} = (0,2)$ supermultiplets together with a $J$-term superpotential $J = \bar{\partial}_A \phi$.

Suppose there is a flavour symmetry $G_F$ acting on the chiral multiplets. Then we can introduce real mass parameters $m \in \mathfrak{t}_F$ by coupling to a background vectormultiplet for $G_F$ and turning on a constant expectation value for the real scalar. As discussed further below, this is complexified by a background Wilson line around $S^1$.

Integrating out charged massive chiral multiplets generates an effective Chern-Simons level $k^{\text{eff}}(\sigma)$ that depends in a piecewise constant fashion on $\sigma$. For $G = U(1)$ and chiral multiplets $\Phi_j$ transforming with weights $Q_j$,

$$k^{\text{eff}}(\sigma) = k + \frac{1}{2}\sum_j Q_j^2 \operatorname{sgn}(Q_j \sigma + m_j), \tag{13}$$

where $m_j$ denote the real mass parameter of $\Phi_j$. In this situation, the bare Chern-Simons level $k$ is allowed to be a half-integer provided $k^{\text{eff}}(\sigma)$ is integer valued. More generally, in the presence of charges chiral multiplets we require that $k^{\text{eff}}(\sigma) \in H^4(BG, \mathbb{Z})$ in order to cancel the parity anomaly.

Finally, the vectormultiplet Lagrangian (8) and chiral multiplet Lagrangian (12) are exact with respect to both of the supercharges $Q$, $\bar{Q}$. On the other hand, the Lagrangians for the supersymmetric Chern-Simons term, FI and real mass parameters are not exact for any combination of these supercharges.

## 2.3 The 1d FI Parameter

The Lagrangian (10) for the 3d FI parameter $\zeta$ is not equal to the standard Lagrangian for a 1d FI parameter due to the relation $D_{1d} := D - 2F_{1\bar{1}}$ between the vectormultiplet auxiliary fields in one and three dimensions. Instead we find

$$L_{\text{FI}} = L_{1,\zeta} - L_{2,\zeta}, \tag{14}$$

where the first term

$$
\begin{aligned}
L_{1,\zeta} &= \frac{i\zeta}{2} \cdot (Q + \bar{Q})(\lambda + \bar{\lambda}) \\
&= -i\zeta \cdot D_{1d}
\end{aligned}
\tag{15}
$$

is the exact Lagrangian of a 1d FI parameter, while the second term

$$L_{2,\zeta} = 2i\zeta \cdot F_{1\bar{1}} \tag{16}$$

is not exact and will weight contributions from different magnetic fluxes on $\Sigma$.

For supersymmetric localisation it is convenient to treat the parameter in $L_{1,\zeta}$ as independent. We will call this parameter $\tau$, and the Lagrangian $L_{1,\tau}$. This is an exact deformation which does not preserve 3d $\mathcal{N} = 2$ supersymmetry, similar to those used in [23–25] to localise the path integral of A-twisted 2d $\mathcal{N} = (2,2)$ gauge theories onto vortex solutions. It will play the same role here in section 4.

From the perspective of supersymmetric quantum mechanics and localisation, $\tau$ and $\zeta$ can be considered to be independent from each other. The former is real and exact, while the latter can be complexified by a Wilson line for the topological symmetry and is not exact. However, to recover 3d $\mathcal{N} = 2$ supersymmetry at the end, we must set $\tau = \zeta$.

## 2.4 Parameter Dependence

As above, it is natural to regard a twisted 3d $\mathcal{N} = 2$ theory as an $\mathcal{N} = (0,2)$ supersymmetric quantum mechanics. From this perspective, the twisted index is identified with the Witten index of this supersymmetric quantum mechanics [8]. Provided the spectrum is gapped, the twisted index can be expressed as

$$\mathcal{I} = \sum_{d \in \pi_1(G)} q^d \operatorname{Tr}_{\mathcal{H}_d} (-1)^F y^{J_F} . \tag{17}$$

In this expression, the summation is over the topological class $d \in \pi_1(G)$ of the principal $G$-bundle on $\Sigma$ and the trace is then over supersymmetric ground states $\mathcal{H}_d$ in each topological sector. The parameters appearing in this expression are

$$y := e^{-2\pi\beta(m + ia_F)}, \qquad q := e^{-2\pi\beta(\zeta + ia_T)}, \tag{18}$$

where $m$, $\zeta$ denote the mass and FI parameters and $a_F$, $a_T$ are background holonomies for the associated global symmetries $G_F$, $G_T$ along $S^1$. The parameters $y$, $q$ are then valued in the complexified maximal tori $T_{F,\mathbb{C}}$, $T_{T,\mathbb{C}}$ and the twisted index is a meromorphic function of them.

Note that the dependence on $q$ arises from the second non-exact contribution to the Lagrangian (14). This contribution, as well as the piece of the Lagrangian containing the real mass $m$, are not exact with respect to any combination of the supercharges generating the $\mathcal{N} = (0,2)$ supersymmetric quantum mechanics and are naturally complexified. The twisted index depends explicitly on $m$, $\zeta$ as a meromorphic function of the complexified parameters $y$, $q$.

In contrast, the Lagrangian $L_\tau$ containing an independent 1d FI parameter $\tau$ is exact with respect to the linear combination $Q + \bar{Q}$. Standard arguments ensure the twisted index is invariant under small deformations of $\tau$, but may jump across codimension-1 walls where the spectrum of the supersymmetric quantum mechanics is not gapped. Importantly, the locations of the walls depend on other parameters of the theory such as $e^2$ and $\text{Vol}(\Sigma)$. This wall-crossing can be studied following the localisation techniques developed for gauged supersymmetric quantum mechanics in [10]. This is the route we follow in section 3.

## 3 Coulomb Branch Localisation

In this section, we reconsider the Jeffrey-Kirwan (JK) contour integral formula for the twisted index, which was derived using the Coulomb branch localisation scheme in [2–4]. We show

that the 1d FI parameter $\tau$ modifies the residue prescription for singularities at the boundary of the moduli space of supersymmetric saddle points in this localisation scheme and provide a general formula for $G = U(1)$.

This modification ensures that the contour integral formula is well-defined and independent of the auxiliary parameter in each individual topological sector $d \in \pi_1(G)$, at least away from codimension-1 walls in the parameter space of $\tau$. This is a pre-requisite for the existence of an effective supersymmetric quantum mechanics whose Witten index captures the contribution from each topological sector; this observation will be important in section 4. Furthermore, it leads to wall-crossing of the twisted index, for which we provide a general formula in the case $G = U(1)$.

### 3.1 Contour Integral Formula

The Coulomb branch localization scheme of [2–4] starts from the Lagrangian

$$L = \frac{1}{e^2} L_{YM} + \frac{1}{g^2} L_{\Phi} + L_{\text{CS}} + L_{\text{FI}}, \tag{19}$$

with parameters $e^2$, $g^2$ multiplying the exact terms. Schematically, one sends $e^2 \to 0$ to localise onto saddle points of the vectormultiplet Lagrangian and then $g^2 \to 0$ to evaluate the contributions from fluctuations of chiral multiplets. This is problematic: additional chiral multiplet zero modes and the non-compactness of the moduli space of vectormultiplet saddles mean that the $e^2 \to 0$ limit is subtle. A careful analysis, following the similar computations in two dimensions [26, 27], leads to a Jeffrey-Kirwan contour integral formula for the twisted index.

The contour integral formula for the twisted index is

$$\mathcal{I} = \frac{1}{|W|} \sum_{\underline{\mathfrak{m}} \in \Lambda_G} q^{\text{Tr}(\underline{\mathfrak{m}})} \int_{\Gamma} \prod_{a=1}^{\text{rk}(G)} \frac{dx_a}{x_a} \, Z(x, \underline{\mathfrak{m}}) \, H(x)^g. \tag{20}$$

The summation is over the cocharacter lattice $\Lambda_G$ of $G$ and $\text{Tr} : \Lambda_G \to \pi_1(G)$ denotes the natural projection onto the fundamental group. The contour integral is in the complexified maximal torus of $G$, parametrised by

$$x = e^{-2\pi\beta(\sigma + i a_0)}. \tag{21}$$

For example, if $G = U(N)$ we have $\underline{\mathfrak{m}} = (\mathfrak{m}_1, \dots, \mathfrak{m}_N) \in \mathbb{Z}^N$ and $\text{Tr}(\underline{\mathfrak{m}}) = \sum_{j=1}^{N} \mathfrak{m}_j$ with Coulomb branch coordinates $x = (x_1, \dots, x_N)$.

Finally, the integrand is constructed from a 1-loop contribution $Z(x, \underline{\mathfrak{m}})$ and the Hessian $H(x)^g$, which arises from integration over the 1-form gaugino zero modes. They also depend on flavour parameters $y$ which are suppressed in the notation.

The computation requires a choice of contour $\Gamma$. As discussed in [2–4] and building on computations in two dimensions [26, 27], the contour is fully determined by zero mode integral over the components of the auxiliary field

$$\hat{D} := i D_{1d} = i(D - 2F_{1\bar{1}}) \tag{22}$$

of the 1d vectormultiplet. The contour for the auxiliary field is given by

$$\Gamma_{\hat{D}} = \mathfrak{t} + i\delta, \tag{23}$$

where the vector $\delta \in \mathfrak{t}$ deforms the contour away from the real slice. After performing the integral over the auxiliary field, the contour $\Gamma$ is given by a JK residue prescription. This

requires choosing an auxiliary parameter $\eta \in \mathfrak{t}^*$, which determines a choice of the vector $\delta$ through relations including $\eta \cdot \delta < 0$.

Let us spell this out for $G = U(1)$. In this case, the JK residue operation is

$$\underset{x=0}{\text{JK-Res}}(Q, \eta)\frac{dx}{x} = \Theta(Q\eta)\text{sgn}(Q), \tag{24}$$

where $\eta \neq 0$ is an auxiliary parameter and $Q \neq 0$ is the JK charge. The the twisted index can then be expressed

$$\mathcal{I} = \sum_{d \in \mathbb{Z}} q^d \sum_{x_*} \underset{x=x_*}{\text{JK-Res}}(Q_{x_*}, \eta)\frac{dx}{x} Z(x, d) H(x)^g, \tag{25}$$

where for $G = U(1)$ the first summation is over $d = \underline{\mathfrak{m}} \in \mathbb{Z}$. The second summation runs over poles $x_* \in \mathbb{C}^*$ of the integrand with JK charges defined as follows.

- First, there are poles at interior points solving equations of the form $x^Q y^{Q_f} = 1$, which arise from chiral multiplets of $U(1)$ charge $Q$ and flavour charge $Q_f$. The associated JK charge is simply $Q$.

- Second, there are poles at the boundary points $x = 0, \infty$, which arise from monopole operators of 't Hooft charge $+1, -1$ and $U(1)$ gauge charge $Q_\pm = \mp k_\pm^{\text{eff}}$ respectively, where we define

$$k_\pm^{\text{eff}} := k^{\text{eff}}(\sigma \to \pm\infty). \tag{26}$$

The associated JK charges are $Q_\pm$.

This is ill-defined as it stands when $k_\pm^{\text{eff}} = 0$. References [2–4] adopt a further regulator such that the pole at $x = 0, \infty$ is not taken when $k_+^{\text{eff}}, k_-^{\text{eff}} = 0$. A consequence is that while the twisted index is independent of $\eta$ after summing over $d \in \mathbb{Z}$, this is not always the case in individual topological sectors. This is not compatible with the existence of an effective supersymmetric quantum mechanics in each topological sector and resolving this issue is one motivation for introducing the parameter $\tau$ below.

There is a similar but more intricate story for non-abelian $G$. At genus $g > 0$, one difficulty is that the integrand of (20) has poles at $x^\alpha = 1$ for non-zero roots $\alpha$, for which the JK residue operation is ill-defined. The prescription adopted in [2–4] is to exclude such poles, which is again problematic for independence of $\eta$ in individual topological sectors. Resolving this issue is beyond the scope of this paper.

## 3.2 Modification due to $\tau$

We now introduce the 1d FI parameter $\tau$ and consider the Lagrangian

$$L = \frac{1}{t^2}\left(\frac{1}{e^2}L_{YM} + L_{1,\tau}\right) + \frac{1}{g^2}L_\Phi + L_{CS} + L_{2,\zeta}, \tag{27}$$

in the limit $t \to 0$ with $e^2$ finite.

We will show below that this changes the residue prescription for the boundary contributions to the twisted index. For simplicity, we focus on $G = U(1)$. In this case, the JK charges associated to the poles at $x = 0, \infty$ become

$$Q_\pm = \begin{cases} \mp k_\pm^{\text{eff}} & \text{if} \quad k_\pm^{\text{eff}} \neq 0 \\ d - \tilde{\tau} & \text{if} \quad k_\pm^{\text{eff}} = 0 \end{cases}, \tag{28}$$

where

$$\tilde{\tau} = \frac{e^2 \text{Vol}(\Sigma)}{2\pi}\tau \tag{29}$$

is a normalised 1d FI parameter. These charges differ from the previous JK residue prescription when $k_\pm^{\text{eff}} = 0$.

Importantly, with the new prescription the charges $Q_\pm$ are always non-vanishing provided $\tilde{\tau} \notin \mathbb{Z}$. Therefore, away from these walls the JK residue prescription is independent of $\eta$ in each individual topological sector $d \in \mathbb{Z}$, before the summation over fluxes. On the other hand, whenever $k_\pm^{\text{eff}} = 0$ it introduces the potential for wall-crossing in the topological sector $d \in \mathbb{Z}$ across the wall $\tilde{\tau} = d$.

The argument follows that outlined in appendix B of [14]. As previously, the boundary contribution is determined by the zero mode integral of the auxiliary field $\hat{D}$. In the new localisation scheme, the boundary contribution to the twisted index is

$$
\begin{aligned}
I_\pm = \sum_{d \in \mathbb{Z}} q^d \lim_{t \to 0} \operatorname*{Res}_{x=0,\infty} \frac{dx}{x} \int_{\mathbb{R}+i\delta} \frac{d\hat{D}}{\hat{D}} \, Z(x,d,\hat{D}) H(x,\hat{D})^g \\
\exp\left[ \frac{\beta \operatorname{vol}(\Sigma)}{2t^2 e^2} \hat{D}^2 - \frac{i\beta}{t^2}\left( -\frac{2\pi d}{e^2} + \operatorname{vol}(\Sigma)\tau + \frac{k_\pm^{\text{eff}}}{2\pi} t^2 \sigma \operatorname{vol}(\Sigma) \right) \hat{D} \right],
\end{aligned}
\tag{30}
$$

where $\delta$ is a regulator for the $\hat{D}$ integral that satisfies $\eta\delta < 0$. The first line of the integrand is the 1-loop and gaugino zero mode contribution in the presence of $\hat{D}$. When evaluated at $\hat{D} = 0$, it reduces to the integrand of (20).

To evaluate this contribution, we need to compute the $\hat{D}$-integral in the limit $t \to 0$ with $t^2\sigma \to \pm\infty$. This integral is performed by rescaling $\hat{D} \to t^2\hat{D}$ such that in the limit $t \to 0$, it is determined by the dominant $\hat{D}$-linear contribution to the exponential. The result can be expressed

$$
I_\pm = \sum_{d \in \mathbb{Z}} q^d \operatorname*{JK\text{-}Res}_{x=0,\infty}(Q_\pm, \eta), \frac{dx}{x} \, Z(x,d) H(x)^g
\tag{31}
$$

where the charge $Q_\pm$ is determined by the dominating contribution to the $\hat{D}$-linear term in the exponential. If $k_\pm^{\text{eff}} \neq 0$, this term dominates and we find $Q_\pm = \mp k_\pm^{\text{eff}}$, in agreement with [2–4]. However, when $k_\pm^{\text{eff}} = 0$ we find instead $Q_\pm = d - \tilde{\tau}$. This is summarised in the residue prescription (28).

## 3.3 Wall-Crossing Formula

The dependence of $Q_\pm$ on $\tau$ leads to wall-crossing of the twisted index when $k_\pm^{\text{eff}} = 0$. Let us again take $G = U(1)$ and consider the change in the twisted index as the normalised 1d parameter crosses an integer value $\tilde{\tau}_* := d_* \in \mathbb{Z}$. A straightforward consequence of the residue prescription (28) is

$$
I(\tau_* - \epsilon) - I(\tau_* + \epsilon) = q^{d_*}\left[ \delta_{k_+^{\text{eff}},0} \operatorname*{Res}_{x=0} + \delta_{k_-^{\text{eff}},0} \operatorname*{Res}_{x=\infty} \right] \frac{dx}{x} Z(x,d_*) H(x)^g,
\tag{32}
$$

where $\epsilon \to 0^+$. We will explore this wall-crossing formula in a large class of examples in section 5 and demonstrate precise agreement with the geometric interpretation of the twisted index to be introduced momentarily in section 4.

## 3.4 Relation to Previous JK Prescription

Let us now use the wall-crossing formula to examine the precise relationship with the original JK residue prescription introduced in [2–4].

The original prescription differs in its treatment of the poles at $x = 0$ and $x = \infty$ when $k_+^{\text{eff}} = 0$ and $k_-^{\text{eff}} = 0$ respectively. In these case, the original prescription is to not include the residue at these points. This prescription is not consistent, meaning not independent of the

auxiliary parameter, in each topological sector $d \in \mathbb{Z}$. However, it is consistent after summing over topological sectors provided

$$\sum_{d \in \mathbb{Z}} q^d \left[ \delta_{k_+^{\mathrm{eff}}, 0} \operatorname*{Res}_{x=0} + \delta_{k_-^{\mathrm{eff}}, 0} \operatorname*{Res}_{x=\infty} \right] \frac{dx}{x} Z(x, d) H(x)^g \tag{33}$$

for $q \neq 1$ is re-summed to zero. An example is that an expression of the form $\sum_{d \in \mathbb{Z}} q^d$ represents a formal delta function and vanishes for $q \neq 1$.

On the other hand, the residue prescription (28) is well defined in every topological sector independently provided $\widetilde{\tau} \notin \mathbb{Z}$. However, the original residue prescription can be recovered by formally sending either $\widetilde{\tau} \to +\infty$ with $\eta > 0$, or $\widetilde{\tau} \to -\infty$ with $\eta < 0$. The equivalence of these two limits amounts to same condition that the sum over topological sectors (33) is re-summed to zero.

We will not write down a wall-crossing formula for a general compact connected group $G$, although this can be studied following techniques from [10]. As mentioned above, one obstacle to doing this systematically is additional poles at $x^\alpha = 1$ for roots $\alpha$. Instead, in section 6 we explore the twisted index of the simplest non-abelian examples with $G = U(N)$ that display wall-crossing phenomena.

# 4 Higgs Branch Localisation

Introducing the 1d FI parameter $\tau$ opens up an alternative localisation scheme leading to an algebro-geometric interpretation of the twisted index as proposed in [14]. In this section, we review how to compute the twisted index in this localisation scheme and explain how wall-crossing arises from change of stability condition in the algebro-geometric context. This is explored further in examples in sections 5 and 6.

## 4.1 Localisation and Saddle Points

We now consider the Lagrangian

$$L = \frac{1}{t^2} \left( \frac{1}{e^2} L_{YM} + L_{1,\tau} + L_\Phi \right) + L_{CS} + L_{2,\zeta}, \tag{34}$$

which is obtained from that used in the Coulomb branch localisation scheme (27) by setting $g = t$. The strategy is to consider the limit $t \to 0$ with $e^2$ fixed. The first step is to enumerate the saddle points in this limit.

Up to boundary terms, the bosonic part of the Lagrangian (34) can be expressed as a sum of complete squares

$$t^2 L \supset \frac{1}{e^2} \left| D + ie^2 \left( \mu(\phi) - \frac{kt^2 \sigma}{2\pi} - \tau \right) \right|^2 + \frac{1}{e^2} \left| -2iF_{1\bar{1}} + e^2 \left( \mu(\phi) - \frac{kt^2 \sigma}{2\pi} - \tau \right) \right|^2$$
$$+ \frac{1}{e^2} |D_\mu \sigma|^2 + \frac{1}{e^2} |F_{01}|^2 + 4|D_{\bar{1}} \phi|^2 + |D_0 \phi|^2 + |\sigma \cdot \phi|^2 - \frac{it^2}{2\pi} \zeta \cdot D, \tag{35}$$

where $\mu(\phi) \in \mathfrak{g}^*$ is the moment map for the action of $G$ on the unitary representation $R$. In the limit $t \to 0$, we can ignore $\zeta$ for the purpose of enumerating saddle points. However, we keep the Chern-Simons level $k$ in anticipation of saddle points where $|\sigma|$ becomes large as $t \to 0$. After integrating out the auxiliary field and imposing reality conditions on the physical fields, the saddle points are

$$-2iF_{1\bar{1}} + e^2 \left( \mu(\phi) - \frac{kt^2 \sigma}{2\pi} - \tau \right) = 0, \tag{36}$$
$$D_\mu \sigma = 0, \quad F_{01} = 0, \quad D_{\bar{1}} \phi = 0, \quad D_0 \phi = 0, \quad \sigma \cdot \phi = 0.$$

Equivalently, they are time-independent solutions to

$$*F_A + e^2 \left( \mu(\phi) - \frac{kt^2\sigma}{2\pi} - \tau \right) = 0 \,,$$
$$d_A\sigma = 0 \,, \qquad \bar{\partial}_A\phi = 0 \,, \qquad \sigma \cdot \phi = 0 \,, \tag{37}$$

where we have translated to an index free notation.

These equations admit a rich spectrum of solutions depending on $\tau$ and the choice of theory. For $G = U(1)$ there is a trichotomy of solutions mirroring the three classes of supersymmetric vacua in flat space considered in [11]. We summarise them below.

1. **Vortex Solutions**
   These are solutions where $\sigma$ remains finite in the limit $t \to 0$. The Chern-Simons level $k$ can be omitted from equations (37) leaving

   $$*F_A + e^2 (\mu(\phi) - \tau) = 0 \,, \qquad \bar{\partial}_A\phi = 0 \,, \qquad \sigma \cdot \phi = 0 \,, \tag{38}$$

   with constant $\sigma$. Integrating the first equation over $\Sigma$ leads to a constraint: to avoid solutions where $\phi$ vanishes identically and $\sigma$ can become infinitely large, we require that $\widetilde{\tau} \neq d$ in the topological sector with flux

   $$d = \frac{1}{2\pi} \int_\Sigma F_A \in \mathbb{Z} \,. \tag{39}$$

   This in turn implies $\sigma = 0$ and therefore equations (38) reduce to abelian vortex equations on $\Sigma$.

2. **Topological Solutions.**
   These are solutions where $|\sigma| \to \infty$ such that the combination $\sigma_0 = t^2\sigma$ remains finite and non-zero as $t \to 0$. This requires $\phi = 0$ identically and the $U(1)$ gauge symmetry is unbroken. Integrating out the massive fluctuations of $\phi$ generates a shift $k \to k_\pm^{\mathrm{eff}}$ when $\pm\sigma_0 > 0$. The problem is therefore reduced to

   $$*F_A + e^2 \left( -\frac{k_\pm^{\mathrm{eff}}\sigma_0}{2\pi} - \tau \right) = 0 \,, \qquad \pm\sigma_0 > 0 \,. \tag{40}$$

   Integrating over $\Sigma$, there is a unique solution for $\sigma_0$ provided $\widetilde{\tau} \neq d$ and the further conditions $k_\pm^{\mathrm{eff}} \neq 0$ and $\mathrm{sgn}(k_\pm^{\mathrm{eff}}) = \pm\,\mathrm{sgn}(d - \widetilde{\tau})$ are satisfied.

3. **Coulomb Solutions.**
   If $k_\pm^{\mathrm{eff}} = 0$, there are no topological vacua with $\pm\sigma_0 > 0$. However, a non-compact Coulomb branch parametrised by $\pm\sigma_0 > 0$ then opens up the wall $\widetilde{\tau} = d$.

In summary, in the topological sector $d \in \mathbb{Z}$, there may be vortex and topological solutions for $\widetilde{\tau} \neq d$, while Coulomb solutions can arise on the wall $\widetilde{\tau} = d$. For a general compact connected group $G$, equations (37) admit a rich variety of solutions combining characteristic features of the three classes introduced above.

We can introduce real mass parameters $m \in \mathfrak{t}_F$ for the flavour symmetry $G_F$ acting on the chiral multiplets. This modifies the equation $\sigma \cdot \phi = 0$ to

$$(\sigma + m) \cdot \phi = 0 \,, \tag{41}$$

where it is understood that $\sigma$, $m$ act in the appropriate representations of $G$, $G_F$. This has no effect on the above description of topological and Coulomb solutions where $\phi = 0$, but restricts vortex solutions to configurations that are invariant under the infinitesimal flavour transformation generated by $m$.

## 4.2 Moduli and $\tau$-Dependence

We denote the bosonic moduli space of solutions to equations (37) modulo gauge transformations by $\mathfrak{M}_\tau$. From the perspective of $\mathcal{N} = (0,2)$ supersymmetric quantum mechanics, this moduli space parametrises chiral multiplet zero modes. This decomposes as a disjoint union of topologically distinct sectors

$$\mathfrak{M}_\tau = \bigsqcup_{d \in \pi_1(G)} \mathfrak{M}_{\tau,d} \,. \tag{42}$$

The moduli space has an intricate dependence on $\tau$ and may jump discontinuously across codimension one walls in the parameter space $\mathfrak{g}_T$. If a non-compact Coulomb branch opens up on this wall, the twisted index may undergo wall-crossing. We therefore distinguish two types of discontinuity.

- **Type I.** The moduli space $\mathfrak{M}_{\tau,d}$ jumps discontinuously across a wall where a Coulomb branch opens up and the twisted index can undergo wall-crossing. For $G = U(1)$ there is such a discontinuity at $\widetilde{\tau} = d$ when $k_+^{\mathrm{eff}} = 0$ or $k_-^{\mathrm{eff}} = 0$ or both.

- **Type II.** The moduli space $\mathfrak{M}_{\tau,d}$ jumps discontinuously without a Coulomb branch opening up and the twisted index remains unchanged. For $G = U(1)$ there is such a discontinuity at $\widetilde{\tau} = d$ when both $k_+^{\mathrm{eff}} \neq 0$ and $k_-^{\mathrm{eff}} \neq 0$.

This is consistent with the wall-crossing formula (32) in predicting when wall-crossing of the twisted index can occur in theories with $G = U(1)$. To illustrate the difference between the two types of discontinuity, we consider a couple of examples.

First consider $G = U(1)$ with a pair of chiral multiplets $X_\pm$ of charge $\pm 1$ and vanishing R-charge. In this case, $k_+^{\mathrm{eff}} = k_-^{\mathrm{eff}} = 0$ so there are no topological solutions, while vortex solutions satisfy

$$*F_A + e^2 \left( |X_+|^2 - |X_-|^2 - \tau \right) = 0 \,, \qquad \bar{\partial}_A X_\pm = 0 \,. \tag{43}$$

Let us assume $d > 0$. If $\widetilde{\tau} > d$, the vortex equations require $X_- = 0$ and $\mathfrak{M}_{\tau,d} = \mathrm{Sym}^d \Sigma$. If $\widetilde{\tau} < d$, the vortex equations have no solutions and $\mathfrak{M}_{\tau,d} = \emptyset$. At $\widetilde{\tau} = d$, a Coulomb branch opens up. We therefore have a type I discontinuity at $\widetilde{\tau} = d$.

Second, consider $G = U(1)$ supersymmetric Chern-Simons theory at level $k \in \mathbb{Z}_{\geq 0} + \frac{1}{2}$ and a chiral multiplet $X$ of charge $+1$ and vanishing R-charge. In this case, $k_\pm^{\mathrm{eff}} = k \pm \frac{1}{2}$ and as usual $k = \frac{1}{2}$ and $k > \frac{1}{2}$ behave differently.

- If $k = \frac{1}{2}$, there are vortex solutions when $\widetilde{\tau} > d$, topological solutions when $\widetilde{\tau} < d$ and a Coulomb branch at $\widetilde{\tau} = d$. The discontinuity is therefore type I.

- If $k > \frac{1}{2}$, there are both vortex and topological solutions when $\widetilde{\tau} > d$, topological solutions only when $\widetilde{\tau} < d$, and no Coulomb branch at $\widetilde{\tau} = d$. The disccontinuity is therefore type II.

### 4.3 Algebro-Geometric Construction

In order to match the wall-crossing formula (32) precisely, we should evaluate the twisted index in the current localisation scheme. This leads to an effective $\mathcal{N} = (0,2)$ supersymmetric quantum mechanics for each $d \in \pi_1(G)$, which is schematically a sigma model whose target space $\mathfrak{M}_{\tau,d}$ parametrises chiral multiplet zero modes. The contribution to the twisted index is captured by the Witten index of this supersymmetric quantum mechanics: schematically the index of a Dirac operator on $\mathfrak{M}_{\tau,d}$.

To make this precise, it is useful to introduce an algebraic description of the moduli space $\mathfrak{M}_{d,\tau}$ as parametrising the following data:

- A holomorphic $G$-bundle $E$ of degree $d \in \pi_1(G)$.

- A holomorphic sections of the associated bundle $K_\Sigma^{r/2} \otimes E_R$.

This is supplemented by '$\tau$-stability', which depends in a piecewise constant fashion on $\tau$. From an algebraic perspective, the discontinuities across walls in the parameter space of $\tau$ arise from a change of this stability condition.

Let us first assume $\tau$ is chosen such that there are only vortex saddle points where $G$ is completely broken and the moduli space $\mathfrak{M}_{\tau,d}$ is smooth. Then the effective supersymmetric quantum mechanics is a sigma model, with target space $\mathfrak{M}_{\tau,d}$ parametrising the $\mathcal{N} = (0,2)$ chiral multiplet zero modes. The contribution to the twisted index is

$$q^d \int \hat{A}(\mathfrak{M}_{\tau,d}) \, \mathrm{Ch}(\mathcal{E}_{\tau,d}), \tag{44}$$

where $\mathcal{E}_{\tau,d}$ is a complex of coherent sheaves on $\mathfrak{M}_{\tau,d}$ encoding $\mathcal{N} = (0,2)$ Fermi multiplet zero modes and supersymmetric Chern-Simons terms. The examples presented in section 5 are of this type.

More generally, the gauge group may not be completely broken at points on $\mathfrak{M}_{\tau,d}$ and it should be understood as an algebraic stack. For example, for topological solutions in theories with $G = U(1)$ it is the Picard stack of holomorphic line bundles on $\Sigma$ of degree $d$. Nevertheless, the supersymmetric field theory equips the moduli space $\mathfrak{M}_{\tau,d}$ with a perfect obstruction theory and equation (44) must be understood in a virtual sense. This more general setup is studied in [20].

Finally, in the presence of mass parameters $m \in \mathfrak{t}_F$ everything should be understood equivariantly with respect to $G_F$. If the moduli space $\mathfrak{M}_{\tau,d}$ is non-compact away from walls where a Coulomb branch opens up, it is necessary to turn on such mass parameters. In such cases, provided the fixed locus of the infinitesimal $T_F$ transformation generated by $m$ is compact, equation (44) is defined by equivariant localisation to this fixed locus.

# 5 Abelian $\mathcal{N} = 4$ Theories

In this section, we explore the wall-crossing formula (32) for 3d $\mathcal{N} = 4$ supersymmetric QED with $N$ hypermultiplets and demonstrate a precise match with wall-crossing in the algebro-geometric construction summarised in section 4, expanding on the proposal of [14]. [1]

## 5.1 Twisted Index

From a 3d $\mathcal{N} = 2$ perspective, we have $G = U(1)$ and chiral multiplets transforming in the following representations,

$$
\begin{array}{c|cccc}
 & U(1)_R & G & PSU(N)_F & 2U(1)_t \\
\hline
X & r & 1 & \bar{N} & 1 \\
Y & r & -1 & N & 1 \\
\Phi & 2-2r & 0 & 1 & -2
\end{array}
\,, \tag{45}
$$

where the final two columns are the representations under the $\mathcal{N} = 2$ flavour symmetry $G_F = PSU(N)_F \times U(1)_t$. The topological symmetry is $G_T = U(1)$.

---

[1]The analysis of this section can be generalised to abelian quiver gauge theories that have isolated massive vacua in the presence of generic mass and FI parameters. See [14] for more detail.

In the above table, $r = 1,0$ denotes a choice of integer $\mathcal{N} = 2$ R-symmetry inside the $\mathcal{N} = 4$ R-symmetry $SU(2)_H \times SU(2)_C$. If the Cartan generators of each factor in the $\mathcal{N} = 4$ R-symmetry are $T_H, T_C$, then the integer $\mathcal{N} = 2$ R-symmetry $U(1)_R$ is generated by $2T_H, 2T_C$ for $r = 1,0$ respectively. The remaining independent combination $T_H - T_C$ generates the $\mathcal{N} = 2$ flavour symmetry $U(1)_t$.

The two choices $r = 1,0$ generate distinct twisted theories on $S^1 \times \Sigma$, which we refer to the H-twist and C-twist respectively. In both cases, the contribution to the twisted index can be expressed in the standard form,

$$\mathcal{I} = \sum_{\mathfrak{m} \in \mathbb{Z}} q^{\mathfrak{m}} \sum_{x_*} \underset{x=x_*}{\text{JK-Res}}(Q_{x_*}, \eta) \frac{dx}{x} Z(x,d) H(x)^g, \tag{46}$$

where

$$Z(x,d) = (t^{1/2} - t^{-1/2})^{(1-2r)(1-g)} \prod_{j=1}^{N} \left( \frac{xy_j^{-1} - t^{1/2}}{1 - xy_j^{-1}t^{1/2}} \right)^d \left[ \frac{xy_j^{-1}t^{1/2}}{(1 - xy_j^{-1}t^{1/2})(xy_j^{-1} - t^{1/2})} \right]^{(1-r)(1-g)} \tag{47}$$

and

$$H(x) = \frac{1}{2} \sum_{j=1}^{N} \left( \frac{1 + xy_j^{-1}t^{1/2}}{1 - xy_j^{-1}t^{1/2}} + \frac{1 + x^{-1}y_jt^{1/2}}{1 - x^{-1}y_jt^{1/2}} \right) \tag{48}$$

and for convenience we have introduced a shift $q \to (-1)^N q$. The fugacities $y_1, \ldots, y_N, t$ obey $\prod_{j=1}^{N} y_j = 1$ and parametrise the complexified maximal torus of the $\mathcal{N} = 2$ flavour symmetry.

We now specify the residue prescription. First, there are poles at $x = y_j t^{-1/2}, y_j t^{1/2}$ for all $j = 1, \ldots, N$ arising from the chiral multiplets $X, Y$. They are therefore assigned JK charges $+1, -1$ respectively. Second, since the effective Chern-Simons level vanishes identically with $\mathcal{N} = 4$ supersymmetry, $k_+^{\text{eff}} = k_-^{\text{eff}} = 0$ and the poles at $x \to 0, \infty$ are assigned charge $Q_\pm = d - \tilde{\tau}$. This residue prescription can be summarised as follows:

- $\eta > 0$: sum the residues at $x = t^{-1/2}y_j$ for all $j = 1, \ldots, N$, together with the residues at $x = 0$ and $x = \infty$ if $\tilde{\tau} < d$.

- $\eta < 0$: sum minus the residues at $x = t^{1/2}y_j$ for all $j = 1, \ldots, N$, together with minus the residues at $x = 0$ and $x = \infty$ if $\tilde{\tau} > d$.

These two choices are equivalent away from $\tilde{\tau} = d$ by Cauchy's theorem and the residue prescription is independent of the auxiliary parameter $\eta$ for each $d \in \mathbb{Z}$.

The twisted index can potentially now jump across the wall at $\tilde{\tau}_* := d_*$ according to the formula

$$\mathcal{I}(\tilde{\tau}_* - \epsilon) - \mathcal{I}(\tilde{\tau}_* + \epsilon) = q^{d_*} \left[ \underset{x=0}{\text{Res}} + \underset{x=\infty}{\text{Res}} \right] \frac{dx}{x} Z(x, d_*) H(x)^g. \tag{49}$$

with $\epsilon \to 0^+$. We must therefore evaluate the residues at $x \to 0, \infty$. First note that due to a cancelations between the the two chiral multiplets, the Hessian $H(x)$ has a simple zero as $x \to 0, \infty$. Combining with the behaviour of the 1-loop determinant

$$Z(x,d) \sim \begin{cases} \mathcal{O}(x^{+Nr(1-g)}) & x \to 0 \\ \mathcal{O}(x^{-Nr(1-g)}) & x \to \infty \end{cases}, \tag{50}$$

we can draw the following conclusions:

- **H-twist** ($r = 1$): There is no wall-crossing for $g > 0$. For $g = 0$, we find the following closed formula for wall-crossing of the twisted index,

$$\mathcal{I}(\widetilde{\tau}_* - \epsilon) - \mathcal{I}(\widetilde{\tau}_* + \epsilon) = (-1)^{Nd} q^{d_*} \frac{t^{Nd_*/2} - t^{-Nd_*/2}}{t^{1/2} - t^{-1/2}}, \tag{51}$$

  where $\epsilon \to 0^+$. The appearance of the Hirzebruch genus of the complex projective space $\mathbb{P}^{|Nd_*|-1}$ can be understood from the algebro-geometric interpretation of the twisted index discussed below.

- **C-twist** ($r = 0$). There is no wall-crossing for $g = 0$ and $g = 1$. For $g > 1$, there is wall-crossing if $N > 1$. We look at some individual cases below.

## 5.2 Geometric Picture

We now show that the wall-crossing formula agrees with the algebro-geometric interpretation of the twisted index summarised in section 4. In the case of $\mathcal{N} = 4$ supersymmetry, the algebro-geometric interpretation of the twisted index was studied in our previous paper [14], to which we refer the reader for further background.

First note that $k^{\text{eff}}(\sigma) = 0$ identically so there are no topological saddle points in the localisation scheme of section 4. The vortex saddle points are solutions to the equations

$$\begin{aligned}
\frac{1}{e^2} * F + \sum_{j=1}^{N} (|X_j|^2 - |Y_j|^2) - \tau &= 0, \\
\bar{\partial}_A X_i = \bar{\partial}_A Y_i = 0, \qquad \sum_{j=1}^{N} X_j Y_j &= 0, \\
d\sigma = 0, \qquad \sigma \cdot X_i = \sigma \cdot Y_i &= 0,
\end{aligned} \tag{52}$$

for all $i = 1, \ldots, N$, modulo the $U(1)$ gauge transformation. The moduli space of solutions decomposes into topologically distinct sectors

$$\mathfrak{M}_\tau = \bigsqcup_{d \in \mathbb{Z}} \mathfrak{M}_{\tau,d}, \tag{53}$$

where $d \in \pi_1(U(1)) = \mathbb{Z}$ is the degree of the gauge bundle on $\Sigma$.

Provided $\widetilde{\tau} \neq d$, we have $\sigma = 0$ and the moduli space $\mathfrak{M}_{\tau,d}$ has an algebraic description parametrising the data:

- a holomorphic line bundle $L$ of degree $d$,

- $N$ holomorphic sections $X_j \in H^0(L \otimes K_\Sigma^{r/2})$ and $N$ holomorphic sections $Y_j \in H^0(L^{-1} \otimes K_\Sigma^{r/2})$ satisfying the constraint $\sum_{j=1}^{N} X_j Y_j = 0$,

supplemented by a stability condition arising from the top equation of (52). The latter depends in a piecewise constant fashion on $\tau$. This is the moduli space of '$\tau$-stable' twisted quasi-maps to the Higgs branch, $\mathcal{M}_H = T^* \mathbb{CP}^{N-1}$.

However, as discussed in section 4.2, when $\widetilde{\tau} = d \in \mathbb{Z}$ there are saddle points where $X_j = Y_j = 0$ for all $j = 1, \ldots, N$ and a non-compact Coulomb branch opens up. The stability condition for the algebraic description of $\mathfrak{M}_{\tau,d}$ can jump across this wall, which is the source of wall-crossing of the twisted index in this localisation scheme.

Let us return to computing the contribution to the twisted index for $\widetilde{\tau} \neq d$. The moduli space $\mathfrak{M}_{\tau,d}$ is not necessarily compact, so the evaluation of the twisted index is problematic.

This is remedied by turning on real mass parameters $m_1, \ldots, m_N$ and $m_t$ valued in a Cartan subalgebra $T_F$ of the $\mathcal{N} = 2$ flavour symmetry. This modifies the bottom line of (52) to

$$\left(\sigma - m_j + \frac{m_t}{2}\right) X_j = 0, \qquad \left(-\sigma + m_j + \frac{m_t}{2}\right) Y_j = 0. \tag{54}$$

The outcome is that vortex saddle points now must now be invariant under the flavour transformation generated by $m_1, \ldots, m_N$ and $m_t$. For generic masses, the moduli space of such configurations is compact.

The moduli space of vortex solutions in the presence of generic mass parameters has an algebraic description as the fixed locus of the induced $T_{F,\mathbb{C}}$-action on $\mathfrak{M}_{\tau,d}$. This is straightforward to evaluate explicitly. Let us define $d_{\pm} := \pm d + r(g-1)$. Then we find that

$$\mathfrak{M}_{\tau,d}^{\text{fixed}} = \bigsqcup_{i=1}^{N} \mathfrak{M}_{d,i} \tag{55}$$

where

$$
\begin{aligned}
\mathfrak{M}_{d,i} &= \begin{cases} \text{Sym}^{d_+} \Sigma, & d_+ \geq 0 \\ \emptyset & d_+ < 0 \end{cases} \quad \text{if} \quad \widetilde{\tau} > d, \\
\mathfrak{M}_{d,i} &= \begin{cases} \text{Sym}^{d_-} \Sigma, & d_- \geq 0 \\ \emptyset & d_- < 0 \end{cases} \quad \text{if} \quad \widetilde{\tau} < d.
\end{aligned}
\tag{56}
$$

We can see here clearly that the moduli space can jump accross the wall $\widetilde{\tau} = d$.

The contribution to the twisted index from $\mathfrak{M}_{\tau,d}$ is then expressed via equivariant localisation as a sum of contributions from each component of the fixed locus

$$q^d \sum_{i=1}^{N} \int_{\mathfrak{M}_{d,i}} \frac{\hat{A}(\mathfrak{M}_{d,i})}{\text{Ch}\left(\widehat{\wedge}^{\bullet} N_{d,i}^{\vee}\right)}, \tag{57}$$

where $N_{d,i}$ denotes a virtual normal bundle to $\mathfrak{M}_{d,i}$ arising from the fluctuations of massive chiral and Fermi multiplets in the $\mathcal{N} = (0,2)$ supersymmetric quantum mechanics, the details of which depend on $r = 1, 0$. In either case, this can be interpreted as a virtual equivariant Euler characteristic (or rather index of Dirac operator) of $\mathfrak{M}_{\tau,d}$ defined via virtual localisation.

The contributions to the integral (57) can be evaluated explicitly using intersection theory on symmetric products and converted into a contour integral using techniques from [18, 28]. The relevant computations are performed in [14]. The result is that this contribution can be expressed as

$$q^d \int_{\Gamma} \frac{dx}{x} Z(x,d) H(x)^g, \tag{58}$$

where the contour $\Gamma$ is given by

- $\widetilde{\tau} > d$: evaluate the residues at $x = t^{-1/2} y_j$, $j = 1, \ldots, N$

- $\widetilde{\tau} < d$: evaluate minus the residues at $x = t^{1/2} y_j$, $j = 1, \ldots, N$

This coincides with the JK residue prescription of section 5.1, where the auxiliary parameter is chosen such that $\text{sign}(\eta) = \text{sign}(\tau - d)$. It therefore correctly reproduces the wall-crossing of the twisted index. We now present some interesting features in each twist.

## 5.3 H-twist

We have already observed that there is no wall-crossing of the twisted index for $g > 0$. Correspondingly, while the moduli space $\mathfrak{M}_{d-\widetilde{\tau}}$ jumps discontinuously across the wall $\widetilde{\tau} = d$, its contribution to the twisted index (57) is unchanged

It is interesting to turn the problem around and ask which fluxes $d$ contribute to the twisted index for a given $\tau$. The description of the fixed locus of the total moduli space $\mathfrak{M}_\tau$ splits into three characteristic regions:

(i) $\widetilde{\tau} > g - 1$

$$\mathfrak{M}_\tau^{\text{fixed}} = \bigsqcup_{I=1}^{N} \bigsqcup_{d=1-g}^{\lfloor \widetilde{\tau} \rfloor} \text{Sym}^{d+g-1} \Sigma. \tag{59}$$

(ii) $1 - g < \widetilde{\tau} < g - 1$

$$\mathfrak{M}_\tau^{\text{fixed}} = \bigsqcup_{I=1}^{N} \bigsqcup_{d=1-g}^{\lfloor \widetilde{\tau} \rfloor} \text{Sym}^{d+g-1} \Sigma \; \sqcup \; \bigsqcup_{I=1}^{N} \bigsqcup_{d=\lfloor \widetilde{\tau} \rfloor + 1}^{g-1} \text{Sym}^{-d+g-1} \Sigma. \tag{60}$$

(iii) $\widetilde{\tau} < 1 - g$

$$\mathfrak{M}_\tau^{\text{fixed}} = \bigsqcup_{I=1}^{N} \bigsqcup_{d=\lfloor \widetilde{\tau} \rfloor + 1}^{g-1} \text{Sym}^{-d+g-1} \Sigma. \tag{61}$$

In region (i) it is clear that the twisted index vanishes for $d < 1 - g$ since the moduli space is empty. In region (ii) it is similarly clear that the twisted index vanishes for $d > g - 1$. Since the twisted index is invariant under wall-crossing, this is true for any $\tau$. We therefore conclude that when $g > 0$ the twisted index truncates to a finite Laurent polynomial in $q$ supported in degrees $1 - g < d < g - 1$. This is indeed the case.

When $g = 0$, the moduli space $\mathfrak{M}_\tau$ in the absence of mass parameters has an explicit description as a disjoint union of projective spaces:

(i) $\widetilde{\tau} > 0$

$$\mathfrak{M}_\tau = \bigsqcup_{d=1}^{\lfloor \widetilde{\tau} \rfloor} \mathbb{P}^{Nd-1}, \tag{62}$$

(ii) $\widetilde{\tau} < 0$

$$\mathfrak{M}_\tau = \bigsqcup_{d=\lfloor \widetilde{\tau} \rfloor + 1}^{-1} \mathbb{P}^{-Nd-1}. \tag{63}$$

Therefore a component of the moduli space disappears or appears as we vary $\widetilde{\tau}$ from $d_* + \epsilon$ to $d_* - \epsilon$. This should reproduce the wall-crossing formula (51). To see this, we note that the twisted index computes the generating function of Hirzebruch-genera of the components of the moduli space,

$$\mathcal{I} = \begin{cases} \displaystyle\sum_{d=1}^{\lfloor \widetilde{\tau} \rfloor} q^d \, \widehat{\chi}_t \left( \mathbb{P}^{Nd-1} \right) & \widetilde{\tau} > 0 \\[4mm] \displaystyle\sum_{d=\lfloor \widetilde{\tau} \rfloor + 1}^{-1} q^d \, \widehat{\chi}_t \left( \mathbb{P}^{-Nd-1} \right) & \widetilde{\tau} < 0 \end{cases}, \tag{64}$$

where

$$\hat{\chi}_t(M) := (-1)^{\text{Dim} M} t^{-\frac{\text{Dim} M}{2}} \chi_{-t}(M) \tag{65}$$

denotes the 'symmetrised' Hirzebruch $\chi_y$-genus.

In particular, we find the wall-crossing formula

$$\mathcal{I}(\widetilde{\tau}_* - \epsilon) - \mathcal{I}(\widetilde{\tau}_* + \epsilon) = \begin{cases} -q^{d_*} \, \widehat{\chi}_t \left( \mathbb{P}^{Nd_*-1} \right) & d_* > 0 \\ 0 & d_* = 0 \\ +q^{d_*} \, \widehat{\chi}_t \left( \mathbb{P}^{-Nd_*-1} \right) & d_* < 0 \end{cases}, \tag{66}$$

with $\epsilon \to 0^+$, which agrees with the expression (51) obtained from the contour integral formula.

## 5.4 C-twist

The description of the fixed locus of the total moduli space $\mathfrak{M}_\tau$ can again be split into two characteristic regions:

(i) $\widetilde{\tau} > 0$

$$\mathfrak{M}_\tau^{\text{fixed}} = \bigsqcup_{i=1}^{N} \bigsqcup_{d=0}^{\lfloor \widetilde{\tau} \rfloor} \text{Sym}^d \Sigma, \tag{67}$$

(ii) $\widetilde{\tau} < 0$

$$\mathfrak{M}_\tau^{\text{fixed}} = \bigsqcup_{i=1}^{N} \bigsqcup_{d=\lfloor \widetilde{\tau} \rfloor+1}^{0} \text{Sym}^{-d} \Sigma. \tag{68}$$

The contribution (57) from each topological sector can be converted to a contour integral in full agreement with (46). However, without performing any computations, we see immediately that the twisted index may only receive contributions from powers $q^d$ with $d \geq 0$ when $\tau > 0$ and $d \leq 0$ when $\tau < 0$. Since the twisted index can only undergo wall-crossing if $g > 1$ and $N > 1$, this implies that in all other cases there is only a non-vanishing contribution proportional to $q^0$. This is indeed the case.

It is also interesting to note that in the limit $t \to 1$, the twisted index should reproduce the Rozansky-Witten invariant of the Higgs branch $\mathcal{M}_H$ in the same chamber as $\tau$ [14]. Correspondingly, in this limit the contributions from $d \neq 0$ cancel for any genus $g$ and there is no wall-crossing.

## 6 A Non-Abelian Example

In this section, we study non-abelian examples with $G = U(N)$ that exhibit wall-crossing phenomena. The relevant moduli spaces $\mathfrak{M}_\tau$ in such theories have an algebraic description parametrising pairs $(E, \phi)$ consisting of a rank $N$ holomorphic vector bundle $E$ and a holomorphic section $\phi$. These data are subject to a stability condition known as '$\tau$-stability' [16–18]. We will also encounter the moduli space $\mathfrak{M}_\tau(\Lambda)$ of $\tau$-stable pairs with fixed determinant $\text{Det}(E) = \Lambda$. These moduli spaces have been studied extensively [18,19,21]. In particular, the Poincaré polynomial of $\mathfrak{M}_\tau$ was computed in [18] for $N = 2$ and the Hodge polynomials for $N = 2, 3$ were computed in [22,29].

As discussed in section 4, the twisted index will compute the holomorphic Euler characteristic of $\mathfrak{M}_\tau$, valued in a holomorphic vector bundle $\mathcal{E}$. For concreteness, we construct supersymmetric gauge theories that gives rise to $\mathcal{E} = \mathcal{O}$ and $\mathcal{E} = T^*\mathfrak{M}_\tau$. The second case, which arises from a theory with $\mathcal{N} = 4$ supersymmetry, computes a symmetrised version of the Hirzebruch $\chi_y$-genus. Applied to such theories, the contour integral formula from section 3 provides an alternative method to compute such geometric invariants and their wall-crossing formulae.

## 6.1 Stable Pairs

Since they are perhaps more unfamiliar than moduli spaces of abelian vortices, we begin with a short review of the moduli spaces of $\tau$-stable pairs from an algebraic perspective. How these moduli spaces arise from supersymmetric gauge theory is discussed in subsequent sections.

We focus for concreteness on $N = 2$. Let us consider a rank-2 holomorphic vector bundle $E$ over $\Sigma$ with fixed determinant $\wedge^2 E = \Lambda$, where $\Lambda$ is a holomorphic line bundle of degree $d > 0$. In addition, we have a non-zero holomorphic section $\phi \in H^0(E)$. A pair $(E, \phi)$ is $\tau$-semi-stable if for all line subbundles $L \subset E$,

$$\begin{aligned} d - \deg(L) &\geq \widetilde{\tau} && \text{if } \phi \in H^0(\Sigma, L), \\ \text{and} \qquad \deg(L) &\leq \widetilde{\tau} && \text{if } \phi \notin H^0(\Sigma, L), \end{aligned} \tag{69}$$

where we define $\widetilde{\tau}$ as in the abelian case (29). A pair is $\tau$-stable if these inequalities hold strictly. We denote the moduli space of such stable pairs by $\mathfrak{M}_\tau(\Lambda)$. The correspondence between stable pairs and vortices is summarised in appendix A.

Let us assume $d > 0$. The moduli space $\mathfrak{M}_\tau(\Lambda)$ is non-empty if and only if

$$\frac{d}{2} < \widetilde{\tau} < d. \tag{70}$$

Furthermore, the $\tau$-stability condition is constant and equivalent to $\tau$-semistability within each of the chambers

$$\max\left(\frac{d}{2}, d - i - 1\right) < \widetilde{\tau} < d - i, \tag{71}$$

labelled by integers $i \in \{0, 1, \ldots, \frac{d-1}{2}\}$. We can therefore reasonably introduce the notation $\mathfrak{M}_i(\Lambda)$ for the moduli space of $\tau$-stable pairs in each of these chambers.[2] As $\widetilde{\tau}$ crosses the integer values in $\frac{d}{2} \leq \widetilde{\tau} \leq d$, the moduli space jumps. We summarise the chamber structure in Figure 1.

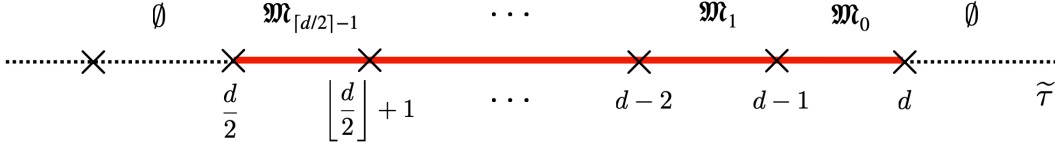

Figure 1: Chamber structure of the moduli space of rank 2 stable pairs. The points marked with $\times$ are the walls across which the description of the moduli space changes.

It is possible to give an explicit algebraic description of $\mathfrak{M}_0(\Lambda)$. First, a consequence of the definition is that stable pairs in $\mathfrak{M}_0(\Lambda)$ are equivalent to non-split extensions of $\mathcal{O}$ by $\Lambda$. In other words, they are holomorphic vector bundles $E$ that fit into a sequence

$$0 \to \mathcal{O} \to E \to \Lambda \to 0 \tag{72}$$

and are not of the form $E \cong \Lambda \oplus \mathcal{O}$, together with a section $\phi \in H^0(\mathcal{O})$. The space of all extensions is $H^1(\mathcal{O} \otimes \Lambda^{-1})$. Removing the split extensions (whose cohomology class is zero) and taking the quotient by the automorphisms gives

$$\mathfrak{M}_0(\Lambda) \cong \mathbb{P}H^1(\Lambda^{-1}). \tag{73}$$

---

[2]There is a similar chamber structure for stable pairs with the determinant unfixed but $c_1(E) = d$ and we denote the corresponding moduli spaces by $\mathfrak{M}_{i,d}$.

More broadly, the moduli spaces $\mathfrak{M}_i(\Lambda)$ are projective but an explicit description is not available for $i > 0$ and other techniques must be exploited. For example, it is possible to understand wall-crossing at the critical values of $\tilde{\tau}$ in terms of Mori theory: when jumping from $\mathfrak{M}_i(\Lambda)$ to $\mathfrak{M}_{i+1}(\Lambda)$, the moduli space undergoes a *flip* where an embedded subvariety is blown-up and the exceptional divisor is blown-down in another direction. With a complete understanding of $\mathfrak{M}_0(\Lambda)$, wall-crossing formulae are then sufficient for computations in other chambers.

For example, the Hodge polynomials of the moduli spaces $\mathfrak{M}_i(\Lambda)$ for $N = 2$ are computed by this method in [22], with the result

$$e(\mathfrak{M}_i(\Lambda)) = \text{coeff}_{x^0}\left( \frac{(1+vx)^g(1+wx)^g}{(1-x)(1-vwx)x^i}\left( \frac{(vw)^i}{1-(vw)^{-1}x} - \frac{(vw)^{d+g-1-2i}}{1-(vw)^2x} \right) \right). \tag{74}$$

The Hirzebruch genus is a found by substituting $v \to -1$, $w \to -t$ in the above expression,

$$\chi_{-t}(\mathfrak{M}_i(\Lambda)) = \text{coeff}_{x^0}\left( \frac{(1-x)^{g-1}(1-tx)^{g-1}}{x^i}\left( \frac{t^i}{1-t^{-1}x} - \frac{t^{d+g-1-2i}}{1-t^2x} \right) \right). \tag{75}$$

Finally, if we do not fix the determinant line bundle $\Lambda$, the moduli space $\mathfrak{M}_{i,d}$ fibres over the Picard variety $\text{Pic}^d(\Sigma)$ parametrising $\Lambda$. The Hodge polynomial is obtained by inserting an additional factor associated to the base,

$$e(\mathfrak{M}_{i,d}) = \text{coeff}_{x^0}\left( \frac{(1+v)^g(1+w)^g(1+vx)^g(1+wx)^g}{(1-x)(1-vwx)x^i}\left( \frac{(vw)^i}{1-(vw)^{-1}x} - \frac{(vw)^{d+g-1-2i}}{1-(vw)^2x} \right) \right). \tag{76}$$

This vanishes in the limit $v \to -1$ and therefore $\chi_{-t}(\mathfrak{M}_{i,d}) = 0$.

Finally, it will be important to give a description of the tangent space to $\mathfrak{M}_\tau(\Lambda)$ as a deformation-obstruction complex: given a stable pair $(E, \phi)$ there is an exact sequence

$$0 \to H^0(\text{End}E) \to H^0(E) \to T_{(E,\phi)}\mathfrak{M}_\tau(\Lambda) \to H^1(\text{End}_0E) \to H^1(E) \to 0, \tag{77}$$

where the maps are induced by multiplication by $\phi$. Here $\text{End}E$ is the bundle of endomorphisms of $E$, while $\text{End}_0E$ is the bundle of trace-free endomorphisms. Colloquially, $H^0(E)$ corresponds to deformations of the section $\phi$, while $H^0(\text{End}E)$ accounts for gauge transformations acting on $\phi$. Similarly, $H^1(\text{End}_0E)$ corresponds to the deformations of the bundle $E$, while $H^1(E)$ are obstructions due to the fact that $\phi$ should remain holomorphic. The trace-free condition has its origin in the fact that we are fixing the determinant of $E$ (trace elements can be identified with the tangent space to the Picard variety parametrizing $\Lambda = \text{Det}E$).

## 6.2 Gauge Theory Construction

We now construct a supersymmetric gauge theory whose twisted index localises to the moduli space $\mathfrak{M}_\tau$ of rank-$N$ stable pairs. A natural candidate is $G = U(N)$ with the following chiral multiplets

|  | $U(2)$ | $U(1)_R$ | $U(1)_t$ |
|---|---|---|---|
| $\mathcal{V}$ | adj | 0 | 0 |
| $\varphi$ | adj | 0 | −1 |
| $X$ | □ | 0 | 0 |
| $Y$ | $\overline{\square}$ | 2 | 1 |

$$\tag{78}$$

and superpotential

$$W = \text{tr}(\varphi XY). \tag{79}$$

This is in fact an $\mathcal{N} = 4$ theory with a particular choice of $\mathcal{N} = 2$ $R$-symmetry.

The effective supersymmetric Chern-Simons levels vanish and vortex like saddle points in the localisation scheme of section 4 are solutions of

$$
\begin{aligned}
&*F_A + e^2 \left( X \cdot X^\dagger - Y^\dagger \cdot Y - 2[\varphi^\dagger, \varphi] - \tau \right) = 0, \\
&\bar{\partial}_A X = \bar{\partial}_A Y = \bar{\partial}_A \varphi = 0, \quad X \cdot Y = \varphi \cdot X = \varphi \cdot Y = 0, \\
&D_A \sigma = 0, \quad \sigma \cdot X = \sigma \cdot Y = [\sigma, \varphi] = 0.
\end{aligned}
\tag{80}
$$

The moduli space of solutions decomposes as a disjoint union of topological sectors labelled by $d \in \pi_1(U(N)) = \mathbb{Z}$. In the following, we focus on $N = 2$.

### 6.2.1 Fixed Loci

We now introduce a real mass parameter $m_t$ for the $U(1)_t$ symmetry, which has the effect of reducing the moduli space to the fixed locus of this symmetry. We show that for $\widetilde{\tau} > d/2$, this coincides with the moduli space of stable pairs $\mathfrak{M}_{\tau,d}$.

First, let us assume that

$$
\widetilde{\tau} > \frac{\deg(E)}{\mathrm{rk}(E)} = \frac{d}{2}.
\tag{81}
$$

Then taking the trace of the first equation in (80) and integrating over $\Sigma$ implies that $X$ is non-vanishing. Then the equation $X \cdot Y = 0$ requires that $Y = 0$ and

$$
\sigma \cdot X = 0 \qquad \varphi \cdot X = 0,
\tag{82}
$$

imply both $\sigma$ and $\varphi$ are generically at most rank one.

Second, let us introduce a mass parameter $m_t$ such that the remaining equations in (80) are modified by the substitution $\sigma \mapsto \sigma + m_t$ and the action in the relevant representation is understood. This leaves the conclusions of the above paragraph unchanged. However, the equations

$$
[\sigma, \varphi] = m_t \varphi \quad D_A \sigma = 0,
\tag{83}
$$

together with the first line of equation (80), imply there are potentially two distinct classes of solutions:

(i) $X \neq 0$, $\varphi = 0$ ($E$ irreducible): Let us first consider solutions with $\varphi = 0$. In this case, as reviewed in appendix A, the bundle $E$ cannot split for generic values of $\tau$. Since $\sigma$ is covariantly constant and $\sigma \cdot X = 0$, it is forced to vanish (otherwise the bundle would split holomorphically). There is therefore a component of the fixed locus defined by the following equations

$$
*F_A + e^2 \left( X \cdot X^\dagger - \tau \right) = 0, \qquad \bar{\partial}_A X = 0,
\tag{84}
$$

with $Y$, $\varphi$ and $\sigma$ all vanishing. This coincides with the moduli space $\mathfrak{M}_{\tau,d}$ of stable pairs of degree $d$.

(ii) $X \neq 0$, $\varphi \neq 0$ ($E$ reducible): Let us now consider solutions with $\varphi \neq 0$. In this case $\sigma$ must have a non-zero constant fixed expectation value and the gauge group is broken to $U(1) \times U(1)$ accordingly. The vector bundle then splits holomorphically,

$$
E \cong L_1 \oplus L_2.
\tag{85}
$$

Without loss of generality, suppose $X \in L_1$. Then

$$
\varphi = \begin{pmatrix} 0 & \varphi_{12} \\ 0 & 0 \end{pmatrix},
\tag{86}
$$

where $\varphi_{12}$ is a section of $L_1 \otimes L_2^{-1}$ of degree $2\mathfrak{m} - d$. Note that the constant expectation value for $\sigma$ is completely fixed by $\sigma \cdot X = 0$ and $[\sigma, \varphi] - m_t \varphi = 0$. We then find two abelian vortex equations

$$
\begin{aligned}
* F_{A_1} + e^2 \left(|X|^2 + 2|\varphi_{12}|^2 - \tau\right) &= 0, \\
* F_{A_2} + e^2 \left(-2|\varphi_{12}|^2 - \tau\right) &= 0,
\end{aligned}
\tag{87}
$$

together with $\bar{\partial}_{A_1} X = \bar{\partial}_{A_1,A_2} \varphi_{12} = 0$. Integrating the second equation over $\Sigma$ shows that solutions exist provided $d - \mathfrak{m} - \tilde{\tau} > 0$. On the other hand, $\varphi_{12}$ can be non-vanishing only when $2\mathfrak{m} - d > 0$ such that the degree of $L_1 \otimes L_2^{-1}$ is non-negative. These two equations imply $\tilde{\tau} < d/2$, which violates our assumption. Therefore this component of the fixed locus is empty.

The moduli space of saddle points in this supersymmetric gauge theory for $\tilde{\tau} > d/2$ can therefore be identified with the moduli space $\mathfrak{M}_{\tau,d}$ of rank 2 stable pairs.

Note that in the opposite region $\tilde{\tau} < d/2$, we find $X = 0$ and the moduli space is instead parametrised by non-vanishing $Y$'s. The first component of the fixed locus with $\varphi = 0$ can be described as solutions to the equations

$$
*F_A + e^2 \left(-Y^\dagger \cdot Y - \tau\right) = 0, \qquad \bar{\partial}_A Y = 0.
\tag{88}
$$

There is a potential second component to the fixed locus with $\varphi \neq 0$, but a similar analysis shows that this is empty for generic $\tau$.

### 6.2.2 Wall-crossing

As discussed in appendix A, at integer values $\tilde{\tau} \in \mathbb{Z}$, the vector bundle $E$ can split holomorphically as (85) even when $\varphi$ vanishes. At these values of $\tilde{\tau}$, the off-diagonal components of the connection vanish identically. As a consequence, $D_A \sigma = 0$ has a family of non-trivial solutions with constant

$$
\sigma = \mathrm{diag}(0, \sigma_2), \qquad \sigma_2 \in \mathbb{R},
\tag{89}
$$

which implies that a non-compact Coulomb branch opens up at this point. Therefore, the supersymmetric observables may jump as $\tilde{\tau}$ varies from $\mathfrak{m} + \epsilon$ to $\mathfrak{m} - \epsilon$ for $\mathfrak{m} \in \mathbb{Z}$ in $d/2 < \mathfrak{m} < d$.

We can derive further constraints directly from the BPS equations. Suppose that the holomorphic section $X$ is contained in the subbundle $L_1$. Then integrating the D-term equation (177) over $\Sigma$ gives the following two relations:

$$
\begin{aligned}
2\pi \mathfrak{m} + ||B||^2 + e^2 \mathrm{vol}(\Sigma) \left(||X||^2 - \tau\right) &= 0, \\
2\pi (d - \mathfrak{m}) - ||B||^2 - e^2 \mathrm{vol}(\Sigma) \tau &= 0,
\end{aligned}
\tag{90}
$$

where $B$ is an off-diagonal component of the gauge connection. First of all, in order to have a non-zero section $X$, $\deg(L_1) = \mathfrak{m}$ must be non-negative. Then the second equation implies that solutions exist only in the sectors $\mathfrak{m} < d - \tilde{\tau}$ and therefore the moduli space is empty when $\tilde{\tau} > d$. If $\tilde{\tau}$ is in the region $(d - i - 1, d - i)$, the equations may admit non-trivial solutions from the sectors $\mathfrak{m} = 0, \cdots, i$.

At the critical point $\tilde{\tau} = d/2$, the BPS equations again admit a non-compact Coulomb branch with [3]

$$
\sigma = \mathrm{diag}(\sigma_1, \sigma_2), \qquad \sigma_1 = \sigma_2 = (\text{constant}) \in \mathbb{R}
\tag{91}
$$

and therefore the index can jump at this point.

---

[3] $B \neq 0$ for these solutions. When $d$ is even, at $\tilde{\tau} = d/2$, there exist additional non-compact branch with $B = 0$ where the vector bundle splits holomorphically.

### 6.2.3 Tangent space

Once we pick a point $p$ on the moduli space $\mathfrak{M}_{\tau,d}$, the massless fluctuations of the 3d $\mathcal{N} = 2$ multiplets decompose into 1d $\mathcal{N} = (0,2)$ multiplets, which organise into the structure of the virtual tangent space around $p$. First of all, fluctuations of the 3d chiral multiplets $(X, Y, \varphi)$ decompose into 1d chiral and Fermi multiplets:

$$
\begin{aligned}
(\delta X, \psi_X) &\in H^0(E_X), & (\eta_X, F_X) &\in H^1(E_X), \\
(\delta Y, \psi_Y) &\in H^0(E_Y), & (\eta_Y, F_Y) &\in H^1(E_Y), \\
(\delta \varphi, \psi_\varphi) &\in H^0(E_\varphi), & (\eta_\varphi, F_\varphi) &\in H^1(E_\varphi),
\end{aligned}
\tag{92}
$$

where $E_\phi = (P \times_G R_\phi) \otimes K_\Sigma^{r_\phi/2}$ is the associated holomorphic vector bundle of the principal gauge bundle $P$ and $R_\phi$ is the representation of the multiplet $\phi = (X, Y, \varphi)$. Explicitly we have

$$
E_X = E, \qquad E_Y = E^* \otimes K_\Sigma, \qquad E_\varphi = \mathrm{End}\, E,
\tag{93}
$$

for the rank-$N$ vector bundle $E = P \times_G M$, where $M$ is the fundamental representation of $U(N)$. Similarly, the 3d $\mathcal{N} = 2$ vector multiplet decomposes into a 1d $\mathcal{N} = (0,2)$ vector multiplet and a chiral multiplet

$$
(\sigma + i a_0, \lambda) \in H^0(\mathrm{End}\, E), \qquad (\delta \bar{A}, \bar{\Lambda}) \in H^1(\mathrm{End}\, E),
\tag{94}
$$

where $H^0(\mathrm{End}\, E)$ parametrises the infinitesimal holomorphic gauge transformations, and $H^1(\mathrm{End}\, E)$ corresponds to the deformation of the vector bundle. Together with massless fluctuations of the $\varphi$ multiplet (the last line of (92)), they form a 1d $\mathcal{N} = (2,2)$ vector and a chiral multiplet respectively.

Note that part of these fluctuations get masses from the Yukawa couplings. One can show that the massless fluctuations correspond to the cohomology of the following pair of complexes [14]:

$$
\begin{aligned}
H^0(\mathrm{End}\, E) &\xrightarrow{\alpha^0} H^0(E_X \oplus E_Y) \xrightarrow{\beta^0} H^1(\mathrm{End}\, E)^* \\
H^1(\mathrm{End}\, E) &\xrightarrow{\alpha^1} H^1(E_X \oplus E_Y) \xrightarrow{\beta^1} H^0(\mathrm{End}\, E)^*.
\end{aligned}
\tag{95}
$$

Here $\alpha^{0,1}$ is the map defined by a multiplication by $(X, -Y)$, and $\beta$ is the map that takes an inner product with $(Y, X)$. The cohomology of the complexes can be identified with the virtual tangent space of the moduli space. Let us focus on the chambers $\widetilde{\tau} > d/N$. On the fixed locus of the $U(1)_t$ symmetry, we have $Y = 0$ and each of the complex splits into two pieces:

$$
\begin{aligned}
H^0(\mathrm{End}\, E) &\xrightarrow{\alpha^0} H^0(E_X), & H^0(E_Y) &\xrightarrow{\beta^0} H^1(\mathrm{End}\, E)^* \\
H^1(\mathrm{End}\, E) &\xrightarrow{\alpha^1} H^1(E_X), & H^0(E_Y) &\xrightarrow{\beta^1} H^0(\mathrm{End}\, E)^*.
\end{aligned}
\tag{96}
$$

One can show that $\alpha^0$ is injective and $\alpha^1$ is surjective when $X$ is non-vanishing. [18] This implies that the cohomology of the two complexes on the left can be identified with the space $T$, which fits into the exact sequence

$$
0 \to H^0(\mathrm{End}\, E) \xrightarrow{\alpha^0} H^0(E) \longrightarrow T \longrightarrow H^1(\mathrm{End}\, E) \xrightarrow{\alpha^1} H^1(E) \longrightarrow 0.
\tag{97}
$$

This coincides with the tangent space of the moduli space of stable pairs, $T = T\mathfrak{M}_{\tau,d}$. On the other hand, from Serre duality the remaining two complexes on the right of (96) can be identified with the shifted cotangent space $-T^*\mathfrak{M}_{\tau,d}$ on the fixed locus.

To summarise, the massless fluctuations give rise to the virtual tangent bundle, which decomposes into

$$T^{\text{vir}}|_{\text{fixed}} = T\mathfrak{M}_{\tau,d} - T^*\mathfrak{M}_{\tau,d} \tag{98}$$

on the fixed locus. The second term is the moving part of the virtual tangent bundle, which has weight $+1$ under the $U(1)_t$ action. The Hilbert space of supersymmetric ground states of the $\mathcal{N} = 2$ quantum mechanics for a fixed degree $d$ is the Dolbeault cohomology valued in the exterior powers of the cotangent bundle:

$$\mathcal{H}_{\tau,d} = \bigoplus_{p,q=1}^{n} H^q\left(\mathfrak{M}_{\tau,d}, \wedge^p t\ T^*\mathfrak{M}_{\tau,d}\right), \tag{99}$$

where $n = \dim_{\mathbb{C}}(\mathfrak{M}_{\tau,d}) = (N^2 - N)(g-1) + d$. The twisted index is therefore a generating function of Hirzebruch-genera of the moduli spaces, [4]

$$\hat{\chi}_t(\mathfrak{M}_{\tau,d}) = \int_{\mathfrak{M}_{\tau,d}} \hat{A}(\mathfrak{M}_{\tau,d}) \text{ch}\left(\widehat{\wedge}^{\bullet} t^{-1} T\mathfrak{M}_{\tau,d}\right), \tag{100}$$

where the 'symmetrised' Hirzebruch genus is defined in equation (65).

The twisted index of the theory (78) with fixed degree $d$ vanishes. This is due to the existence of a fibration over the Jacobian,

$$\mathfrak{M}_{\tau,d} \to \text{Jac}^d[\Sigma], \tag{101}$$

whose fiber is isomorphic to

$$\mathfrak{M}_{\tau}(\Lambda), \tag{102}$$

which is the moduli space of $\tau$-stable pairs with the determinant line bundle $\Lambda$ fixed. Since the Hirzebrich genus of the Jacobian base identically vanishes, a standard argument shows that the twisted index will also vanish[5]

In order to compute something non-trivial from the gauge theory side, we work with the moduli space (102) with the determinant $\Lambda$ fixed in what follows. This can be done by freezing by hand the massless fluctuations corresponding to the direction tangent and co-tangent to the Jacobian. They are generated by the trace of the 1d $\mathcal{N} = (0, 2)$ chiral and Fermi multiplets

$$\left(\text{tr}(\delta\bar{A}), \text{tr}(\bar{\Lambda})\right) \in H^1(\text{End}E) \ \ \text{and} \ \ \text{tr}(\eta_\varphi) \in H^1(\text{End}E)^* \tag{103}$$

respectively. Below we will perform the path integral inserting a delta function that retains only the traceless part of these multiplets. The remaining massless degrees of freedom now corresponds to

$$T^0_{\text{vir}}|_{\text{fixed}} = T\mathfrak{M}_{\tau}(\Lambda) - T^*\mathfrak{M}_{\tau}(\Lambda), \tag{104}$$

where $T\mathfrak{M}_{\tau}(\Lambda)$ fits into the exact sequence (77). Note that this is different from considering the $G = SU(N)$ version of the theory, which would also remove the trace-free part of the 1d $\mathcal{N} = (0, 2)$ vector multiplet contribution $H^0(\text{End}E)$.

## 6.3 Decoupling Limits and Generalisations

It is interesting to consider the limit $|m_t| \to \infty$ where the supermultiplets $Y$ and $\Phi$ charged under the $U(1)_t$ symmetry are integrated out.

---

[4]Here $\widehat{\wedge}^{\bullet}$ is the symmetrised exterior algebra defined by $\widehat{\wedge}^{\bullet}V = (\det V)^{-1/2} \otimes \wedge^{\bullet}V$. The determinant factor is due to the symmetric quantization for fermions. Note that we have $\text{td}(M)\text{ch}(\wedge^{\bullet}TM) = \hat{A}(M)\text{ch}(\widehat{\wedge}^{\bullet}TM)$.

[5]A similar statement is the fact that the twisted index of 3d $\mathcal{N} = 4$ supersymmetric QCD is zero when $N_c < N_f$.

- First, in the limit $m_t \to -\infty$ (corresponding to $t \to \infty$), the top exterior power of the cotangent bundle dominates in the expression (99) and the space of supersymmetric ground states becomes

$$\mathcal{H}_{\tau,d} = H_{\bar{\partial}}^{0,\bullet}\left(\mathfrak{M}_{\tau,d}, \mathcal{K}\right). \tag{105}$$

The fermions in the massive supermultiplets generate the CS-levels [6]

$$\Delta k_{U(1)} = -\frac{1}{2}, \qquad \Delta k_{SU(N)} = N - \frac{1}{2}, \qquad \Delta k_{U(1)R} = \frac{1}{2}, \qquad \Delta k_R = -\frac{1}{2}. \tag{107}$$

- Second, in the opposite limit $m_t \to \infty$ (corresponding to $t \to 0$), the space of supersymmetric ground states become

$$\mathcal{H}_{\tau,d} = H_{\bar{\partial}}^{0,\bullet}\left(\mathfrak{M}_{\tau,d}, \mathcal{O}\right) \tag{108}$$

and integrating out massive fermions generates CS-levels

$$\Delta k_{U(1)} = \frac{1}{2}, \qquad \Delta k_{SU(N)} = -N + \frac{1}{2}, \qquad \Delta k_{U(1)R} = -\frac{1}{2}, \qquad \Delta k_R = \frac{1}{2}. \tag{109}$$

More generally, one can consider an $\mathcal{N} = (0, 2)$ quantum mechanics whose space of supersymmetric ground states can be identified with the cohomology valued in $\mathcal{E} = \mathcal{K}^{p+1/2}$, where $\mathcal{K}$ be the determinant line bundle over $\mathfrak{M}_{\tau,d}$. For this purpose, we consider the following 3d $\mathcal{N} = 2$ theory:

$$\begin{array}{c|c|c} & U(N)_G & U(1)_R \\ \hline \mathcal{V} & \text{adj} & 0 \\ X & \square & 0 \end{array} \tag{110}$$

together with the CS levels

$$k_{U(1)} = -p, \qquad k_{SU(N)} = (2N-1)p, \qquad k_{U(1)R} = p, \qquad k_{RR} = -p, \tag{111}$$

where $k_{U(1)R}$ is the mixed CS level between the $U(1) \subset U(N)_G$ and the $U(1)_R$ R-symmetry. The level should be an half-integer, $p + \frac{1}{2} \in \mathbb{Z}$, such that the line bundle $\mathcal{K}^{p+1/2}$ is well-defined. This is compatible with the above $|m_t| \to \infty$ limits of the $\mathcal{N} = 4$ theory.

As above we work with the moduli space $\mathfrak{M}_\tau(\Lambda)$ with fixed determinant line, by removing the trace of the fluctuation of 1d $\mathcal{N} = (0, 2)$ chiral multiplet

$$\left(\text{tr}(\delta\bar{A}), \text{tr}(\bar{\Lambda})\right) \in H^1(\text{End}E) \tag{112}$$

in the path integral of effective quantum mechanics. As before, we focus on $N = 2$.

### 6.3.1 Branches of the moduli space for rank 2 theories

Unlike the $\mathcal{N} = 4$ theory in section 6.2, the relevant BPS equations in the $\mathcal{N} = 2$ theory with generic CS levels (111) has contribution from non-zero effective levels $k^{\text{eff}}(\sigma)$. Let us first consider the rank 2 case with $\mathcal{E} = \mathcal{O}$, where the bare CS levels are given by

$$k_{U(1)} = \frac{1}{2}, \qquad k_{SU(N)} = -\frac{3}{2}, \qquad k_{U(1)R} = -\frac{1}{2}, \qquad k_R = \frac{1}{2}. \tag{113}$$

If $\tilde{\tau}$ is generic in the region $\tilde{\tau} > \frac{d}{2}$, there are again two branches of solutions:

---

[6] The $SU(N)$ effective levels are obtained from the formula

$$k_{\text{eff},G} = k_G + \sum_R \frac{1}{2} T_2(R)\text{sign}(m_t), \tag{106}$$

where $T_2(R)$ is the quadratic index of the representation $R$, normalised in a way that $T_2(\text{fund.}) = 1$.

(i) $X \neq 0$, $E$ is irreducible: In this branch, the vector bundle bundle $E$ does not split holomorphically and we have $\sigma = 0$ by the argument above (84). Therefore the underlying bosonic moduli space is again given by the rank 2 stable pairs $\mathfrak{M}_\tau(\Lambda)$, parametrised by solutions $(A, X)$ to the equations (84). We call this branch the *stable pair branch* or the *irreducible branch*.

(ii) $X \neq 0$, $E$ is reducible: In this case, the BPS equations have solutions with non-zero $\sigma = \text{diag}(\sigma_1, \sigma_2)$, which breaks the gauge group into $U(2) \rightarrow U(1)_1 \times U(1)_2$. This allows the solution with the vector bundle $E$ splitting holomorphically into $L_1 \oplus L_2$. Without loss of generality, let us assume that $X$ is a non-zero section of $L_1$. Then the equation $\sigma \cdot X = 0$ sets $\sigma_1 = 0$. Since $\sigma_2$ can still be non-zero, a part of the fluctuation of the $X$ multiplet becomes massive and generates the effective CS level for the $U(1)_2$ unbroken gauge symmetry. The semi-classical D-term equation is then [7]

$$*F_A + e^2 \left( XX^\dagger - \frac{k_\pm^{\text{eff}} \sigma}{2\pi} - \tau \right) = 0, \tag{114}$$

where $k_\pm^{\text{eff}} \sigma$ is a diagonal matrix

$$k_\pm^{\text{eff}} \sigma = \begin{pmatrix} \sigma_2 & 0 \\ 0 & -\frac{1}{2}\sigma_2 \pm \frac{1}{2}\sigma_2 \end{pmatrix}, \tag{115}$$

valid in the region $\text{sign}(\sigma_2) = \pm 1$ with $\sigma_1 = 0$. [8] Taking the trace and integrating the equation over $\Sigma$, we find that this branch exists only in the region $\tilde{\tau} > d/2$. In the region $\sigma_2 > 0$, each diagonal component gives the relation

$$2\pi \mathfrak{m} + e^2 \text{vol}(\Sigma) \left( ||X||^2 - \frac{\sigma_2}{2\pi} - \tau \right) = 0,$$
$$2\pi(d - \mathfrak{m}) - \tau e^2 \text{vol}(\Sigma) = 0, \tag{116}$$

where $\mathfrak{m} = \deg(L_1)$. From the second equation, we find that the branch is empty at generic value of $\tilde{\tau}$. On the other hand, in the region $\sigma_2 < 0$, we have

$$2\pi \mathfrak{m} + e^2 \text{vol}(\Sigma) \left( ||X||^2 - \frac{\sigma_2}{2\pi} - \tau \right) = 0,$$
$$2\pi(d - \mathfrak{m}) + e^2 \text{vol}(\Sigma) \left( \frac{\sigma_2}{2\pi} - \tau \right) = 0. \tag{117}$$

Solutions to these equations exist in the region $d - \tilde{\tau} > \mathfrak{m}$. Note that the second equation completely fixes the value of $\sigma_2$. Then the first component of the BPS equation modulo $U(1)$ gauge trasformations reduces to the abelian vortex equation on $\Sigma$, whose moduli space is the $\mathfrak{m}$-th symmetric product of the curve $\Sigma$. The remaining $U(1)$ gauge symmetry is left unbroken.

We call this branch the *reducible branch*.

In constrast to the case with $k_{\text{eff}} = 0$, non-compact Coulomb branch does not appear at $\tilde{\tau} \in \mathbb{Z}$ in this region. In branch (ii) the expectation value of $\sigma$ is completely determined by the BPS equations and the moduli space remains compact. As $\tilde{\tau}$ crosses the integer values in

---

[7]In what follows, we omit the subscript 0 from $\sigma_0$ to avoid clutter in the notation.

[8]The bare CS levels for $U(1)_1 \times U(1)_2$ can be obtained from the expression (113). The $U(2)$ gauge group breaks into $U(1)_A \times U(1)_B$ in this chamber, where $U(1)_B$ is the maximal torus of $SU(2)$ factor and $U(1)_A$ is the $U(1)$ factor in $U(2)$. This gives the relation $U(1)_{1,2} = \frac{1}{2}[U(1)_A \pm U(1)_B]$ respectively. It is straightforward to check that the CS levels in this basis are $k^{11} = k^{22} = \frac{1}{2}(k_{U(1)} + k_{SU(2)})$ and $k^{12} = k^{21} = \frac{1}{2}(k_{U(1)} - k_{SU(2)})$. Since $\sigma_2$ is non-zero, only $k^{22}$ gets a correction after integrating out $X$ multiplet fluctuation. This gives the expression (115).

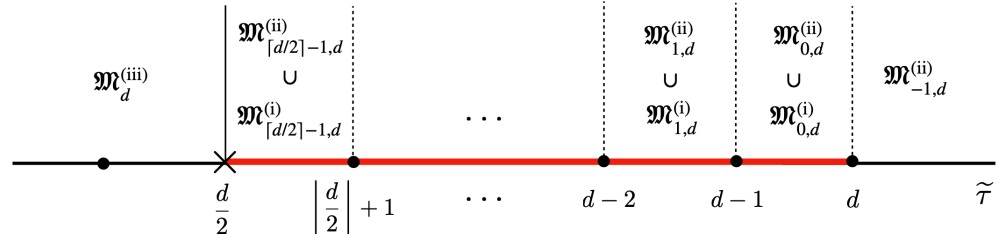

Figure 2: Chamber structure of the theory (110) for $N = 2$, $p = -1/2$. The points denoted by $\times$ are the positions of type I walls, across which the index may jump. The points denoted by black dots are the positions of type II walls. As $\widetilde{\tau}$ crosses the integer values in the red region, the description of the moduli space changes, but the index remains the same. $\mathfrak{M}_{i,d}^{(i)}$, $\mathfrak{M}_{i,d}^{(ii)}$ and $\mathfrak{M}_{i,d}^{(iii)}$ denote for the chambers (i), (ii) and (iii) in the moduli space respectively, where we have $\mathfrak{M}_{i,d}^{(i)} = \mathfrak{M}_{i,d}$, the moduli space of rank 2 stable pairs.

this region, the states associated with branch (i) can appear/disappear according to the wall-crossing phenomena described in section 6.2. However, branch (ii) also can undergo a wall-crossing at the same time, which is expected to compensate the change of the index. Therefore, the wall at $\widetilde{\tau} \in \mathbb{Z}$ is of type II, and the index of the full gauge theory remains constant in this region.

When $\widetilde{\tau}$ is precisely at the critical value $\widetilde{\tau} = \frac{d}{2}$, the equations have non-trivial solutions with $X = 0$. Since $k_{\mathrm{eff}, U(1)} = 0$ for $\sigma_{U(1)} < 0$, the moduli space at this point has a non-compact direction parametrised by negative $\sigma_{U(1)}$. This is a type I wall, where the twisted index can jump. If $\widetilde{\tau}$ goes below the critical value, i.e., $\widetilde{\tau} < \frac{d}{2}$, we encounter another branch:

(iii) $X = 0$: In this branch, the bosonic moduli space is parametrised by solutions $(A, \sigma)$ to

$$*F_A + e^2 \left( -\frac{k^{\mathrm{eff}} \sigma}{2\pi} - \tau \mathbf{1} \right) = 0, \qquad D_A \sigma_0 = 0. \tag{118}$$

The vacua at this branch are described by the Chern-Simons theory with the following effective levels:

$$k_{U(1)}^{\mathrm{eff}} = \begin{cases} 1, & \sigma_{U(1)} > 0 \\ 0, & \sigma_{U(1)} < 0 \end{cases}, \qquad k_{SU(2)}^{\mathrm{eff}} = \begin{cases} -1, & \sigma_{SU(2)} > 0 \\ -2, & \sigma_{SU(2)} < 0 \end{cases}. \tag{119}$$

In the flat space limit, the space of supersymmetric vacua consists of isolated points with a mass gap. We call this branch the *topological branch*.

In Figure 2, we summarise the chamber structure of this theory in the space of $\widetilde{\tau}$.

### 6.3.2 General CS level $|p| > \frac{1}{2}$

The chamber structure of the theory with $p \neq \pm 1/2$ is qualitatively different from that of the case with $p = \pm 1/2$.

First of all, branch (i) remains the same and can be again identified with the moduli space of stable pairs with $\sigma = 0$, which is non-empty in the chambers in $d/2 < \widetilde{\tau} < d$.

In branch (ii), where the vector bundle is reducible, the effective levels are given by

$$k_{\pm}^{\mathrm{eff}} \sigma = \begin{pmatrix} -2p\sigma_2 & 0 \\ 0 & (p \pm 1/2)\sigma_2 \end{pmatrix}. \tag{120}$$

Taking the trace and integrating the D-term equation over $\Sigma$, we find that branch (ii) is potentially non-empty for all values of $\tilde{\tau}$, unlike the case with $p = -\frac{1}{2}$. Integrating each component of the D-term equation now gives

$$2\pi\mathfrak{m} + e^2\text{vol}(\Sigma)\left(||X||^2 + \frac{2p\sigma_2}{2\pi} - \tau\right) = 0,$$
$$2\pi(d-\mathfrak{m}) + e^2\text{vol}(\Sigma)\left(-\frac{(p \pm 1/2)\sigma_2}{2\pi} - \tau\right) = 0,$$

(121)

in the region $\text{sign}(\sigma_2) = \pm 1$ respectively. In this case, we find that solutions exist in the region $p\sigma_2 < 0$ in the sectors $d - \tau < \mathfrak{m}$. On the other hand, in the region $p\sigma_2 > 0$, solutions exist in the sectors $d - \tau > \mathfrak{m}$.

Finally, branch (iii), which corresponds to the topological branch, can be described as solutions to (118) with

$$k_{U(1)}^{\text{eff}} = \begin{cases} -p + \frac{1}{2}, & \sigma_{U(1)} > 0 \\ -p - \frac{1}{2}, & \sigma_{U(1)} < 0 \end{cases}, \qquad k_{SU(2)}^{\text{eff}} = \begin{cases} 3p + \frac{1}{2}, & \sigma_{SU(2)} > 0 \\ 3p - \frac{1}{2}, & \sigma_{SU(2)} < 0 \end{cases}.$$

(122)

By taking the trace of the D-term equations, we find that the solutions can potentially exist for all values of $\tilde{\tau}$.

Note that the value of $\sigma$ is fixed in all regions and therefore all the walls are type II. We expect the twisted index to be independent of $\tau$.

## 6.4 Twisted Index and Wall-Crossing

Now we are ready to discuss the twisted indices of the gauge theories we studied in the last subsection. We propose a generalisation of the wall-crossing formula (32) to the rank-2 theories and show that it reproduces the Hirzebruch genus at each chambers (75). We also compute the twisted indices of the theory (110) and show that it reproduces the holomorphic Euler characteristics valued in a power of the determinant line bundle for the rank 2 stable pairs.

### 6.4.1 $\chi_{-t}(\mathfrak{M}_\tau)$

Twisted index of the theory (78) for $N = 2$ can be written as the integral formula

$$I(\tau) = \sum_{d\in\mathbb{Z}} q^d\, I(\tau, d),$$

(123)

where the summation is over the degree $d \in \mathbb{Z}$ of the principal bundle $P$:

$$I(\tau, d) = \frac{1}{2}\sum_{\substack{(\mathfrak{m}_1,\mathfrak{m}_2)\in\mathbb{Z}^2 \\ \mathfrak{m}_1+\mathfrak{m}_2=d}} \oint_{\text{JK}} \prod_{i=1,2} \frac{dx_i}{x_i}\, Z_{\text{1-loop}}(x_1, x_2, \underline{\mathfrak{m}}) H(x_1, x_2)^g\, + I_{\text{boundary}}(\tau).$$

(124)

As discussed in section 3.1, the first term is the contribution from the contours around the selected poles of the integrand at finite $u_*$, while the second term is the contribution from the boundary of the classical Coulomb branch $|u_i|^2 \to \infty$, which encodes the dependence on the stability parameter $\tau$. The one-loop determinant can be written as

$$Z_{\text{1-loop}} = (-1)^{\mathfrak{m}_1-\mathfrak{m}_2} t^{(-n+g)/2}(1-t)^{-g}(1-t)^{2(g-1)}(x_1-x_2)^{1-g}(x_2-x_1)^{1-g}$$
$$(x_1 - tx_2)^{g-1-\mathfrak{m}_1+\mathfrak{m}_2}(x_2 - tx_1)^{g-1-\mathfrak{m}_2+\mathfrak{m}_1}$$
$$\left(\frac{x_1 - t}{1 - x_1}\right)^{\mathfrak{m}_1+1-g}\left(\frac{x_2 - t}{1 - x_2}\right)^{\mathfrak{m}_2+1-g},$$

(125)

where $n = 2(g-1) + \mathfrak{m}_1 + \mathfrak{m}_2$. Note that we removed the contribution from the the trace of the Fermi multiplet $\text{tr}(\eta_\varphi) \in H^1(\mathcal{O})$ as discussed in (103), by multiplying the factor $(1-t)^{-g}$ in the first line.

The factor $H(x_1, x_2)^g$ in the integrand can be obtained by integrating out the zero modes of the chiral multiplets $(\bar{A}, \bar{\Lambda})$, which correspond to unbroken $U(1)^2 \subset U(2)$ flat connections on $\Sigma$ in the Coulomb branch. Removing the contribution tangent to $\text{Jac}^d[\Sigma]$ in $\mathfrak{M}_{\tau,d}$ can be implemented by inserting the delta function in the field space

$$\delta(\text{tr}(\bar{\Lambda}))\delta(\text{tr}(\bar{A})) \cdot \delta(\text{tr}(\Lambda))\delta(\text{tr}(A)). \tag{126}$$

Using the Lagrange multipliers, it is straightforward to show that, for $G = U(N)$, the result can be written as

$$H(x_1, \cdots, x_N) = \det_{ab}(H_{ab} + \eta)\Big|_{\eta-\text{linear}}, \tag{127}$$

where

$$H_{ab} = \frac{\partial \mathcal{W}_{\text{eff}}(u)}{\partial u_a \partial u_b}, \qquad a = 1, \cdots, N \tag{128}$$

is the hessian of the second derivative of the twisted effective superpotentials of the 3d theory on $S^1$. Here $|_{\eta-\text{linear}}$ means taking the coefficient of the $\eta$-linear term. Explicitly,

$$H_{ab} = \sum_{\alpha \in \Delta} \alpha^a \alpha^b \frac{1 + x^\alpha t^{-1}}{2(1 - x^\alpha t^{-1})} + \delta_{ab} \left[ \frac{1 + x_a}{2(1 - x_a)} + \frac{x_a + t}{2(x_a - t)} \right], \tag{129}$$

where $\Delta$ is the set of all roots in $\mathfrak{g}$. For the rank 2 case, it reads

$$\begin{aligned}
H(x_1, x_2) &= \frac{1 + x_1}{2(1 - x_1)} + \frac{1 + x_2}{2(1 - x_2)} + \frac{x_1 + t}{2(x_1 - t)} + \frac{x_2 + t}{2(x_2 - t)} \\
&\quad + 2\left( \frac{x_2 + x_1 t}{x_2 - x_1 t} + \frac{x_1 + x_2 t}{x_1 - x_2 t} \right).
\end{aligned} \tag{130}$$

Having discussed the integrand, we now turn to the contours. The contours in the expression (124) are determined by the zero mode integrals for the auxiliary field $\hat{D}$, as briefly summarised in section 3.1. If we choose $\eta = (1,1)$, one can show that the only poles in the bulk that pass the JK-residue condition are at $\{x_1 = 1, x_2 = t\}$ and $\{x_2 = 1, x_1 = t\}$. However, the residues of these poles vanish due to the zeros in the numerator of the integrand. Therefore, the contribution from the contour at finite $u$ (the first term of (124)) is identically zero and the index gets contribution from the boundary term only. [9] The index can be written as

$$I(\tau, d) = \sum_{\mathfrak{m}} I^{\mathfrak{m}}_{\text{boundary}}(\tau, d). \tag{131}$$

With this choice of $\eta$, the contour for the $\hat{D}$-integral can be chosen to be (23) with $\delta = (-\epsilon, -\epsilon)$ where $\epsilon$ is small and positive. Then it is possible to show that, on the boundary components of the $x$-contour, the $\hat{D}$-integral has a non-vanishing contribution only around the singularities at

$$\{x_1 = 1, x_2 = 0, \infty\} \text{ and } \{x_2 = 1, x_1 = 0, \infty\}, \tag{132}$$

and all the other contributions vanish after closing the $\hat{D}$ contour. The integrals around these poles depend on the parameter $\tau$. Due to the Weyl symmetry, the contribution from the two singularities are the same after summing over $\mathfrak{m}$.

---

[9]The residue of pole at $x_1 = x_2$ can be non-zero but they are excluded in the prescription as explained in section 3.1

Let us examine the iterative residue integral around the first pole in (132). Using the condition $\mathfrak{m}_1 + \mathfrak{m}_2 = d$, we redefine $(\mathfrak{m}_1, \mathfrak{m}_2) = (\mathfrak{m}, d - \mathfrak{m})$ and sum over $\mathfrak{m} \in \mathbb{Z}$ to obtain the index at a fixed degree $d$. Suppose that $g_{\mathfrak{m}}(x_1, x_2, \hat{D}_1, \hat{D}_2)$ is the product of the one-loop and the classical contributions of the path integral computed around a background with $\hat{D}$ turned on, which reduces to the integrand of (124) when evaluated at $\hat{D} = 0$. After performing the $\hat{D}_1$ integral and taking into account the Weyl symmetry, we are left with the expression

$$I_{\text{boundary}}^{\mathfrak{m}} = \lim_{s \to 0} \left( \operatorname*{res}_{x_2=0} + \operatorname*{res}_{x_2=\infty} \right) \frac{dx_2}{2\pi i x_2} \int_{\mathbb{R}+i\delta_2} \frac{d\hat{D}_2}{\hat{D}_2} \operatorname*{res}_{x_1=1} \frac{dx_1}{x_1} g_{\mathfrak{m}}(x_1, x_2, 0, \hat{D}_2) \, e^{F(s,\hat{D}_2,d-\mathfrak{m},\tau)}, \tag{133}$$

where

$$F(s, \hat{D}_2, d - \mathfrak{m}, \tau) = \frac{\beta \operatorname{vol}(\Sigma)}{2s^2 e^2} \hat{D}_2^2 - \frac{i\beta}{s^2} \left( -\frac{2\pi(d - \mathfrak{m})}{e^2} + \operatorname{vol}(\Sigma)\tau \right) \hat{D}_2. \tag{134}$$

Performing the $\hat{D}_2$ integral after rescaling $\hat{D}_2 \to s^2 \hat{D}_2$, we arrive at the expression

$$I_{\text{boundary}}^{\mathfrak{m}} = \Theta(d - \mathfrak{m} - \widetilde{\tau}) \left( \operatorname*{res}_{x_2=0} + \operatorname*{res}_{x_2=\infty} \right) \operatorname*{res}_{x_1=1} \frac{dx_2}{x_2} \frac{dx_1}{x_1} Z_{\text{1-loop}}(x, \underline{\mathfrak{m}}) \, H(x)^g. \tag{135}$$

Let us fix a generic $\widetilde{\tau}$ in the region $d/2 < \widetilde{\tau} < d$. Note that the pole at $x_1 = 1$ exists only when $\mathfrak{m} \geq 0$. The index can be written as

$$I(\tau, d) = \sum_{\mathfrak{m}=0}^{d-\lfloor \widetilde{\tau} \rfloor - 1} \left( I_{1;0}^{\mathfrak{m}} + I_{1;\infty}^{\mathfrak{m}} \right), \tag{136}$$

where we defined

$$I_{a;b}^{\mathfrak{m}}(d) = \operatorname*{res}_{x_2=b} \operatorname*{res}_{x_1=a} \frac{dx_2}{x_2} \frac{dx_1}{x_1} Z_{\text{1-loop}}(x, \mathfrak{m}) \, H(x)^g. \tag{137}$$

By the residue theorem on the $x_2$-plane with $x_1$ fixed, we can rewrite the index as

$$I(\tau, d) = - \sum_{\mathfrak{m}=0}^{d-\lfloor \widetilde{\tau} \rfloor - 1} I_{1;1}^{\mathfrak{m}}. \tag{138}$$

This expression reflects the fact that the fixed locus under the action of the $U(1)_t$ symmetry is at $\sigma_1 = \sigma_2 = 0$, as discussed in section 6.2. The index (138) is expected to compute the Hirzebruch genus of the moduli space $\mathfrak{M}_{\tau,d}(\Lambda)$ of rank 2 stable pairs in all of the chambers in $d/2 < \widetilde{\tau} < d$. Notice that the bounds agree with the discussion in 6.2.2.

In the chamber $d - 1 < \widetilde{\tau} < d$, the index gets a contribution from the $\mathfrak{m} = 0$ sector only. The corresponding moduli space is the projective space $\mathfrak{M}_0(\Lambda)$ as discussed in section 6.1. In this case the singularity at $x_1 = 1$ is a simple pole and the expression (138) can be explicitly evaluated:

$$\begin{aligned} I(\tau, d) &= - \operatorname*{res}_{x_2=1} \frac{dx_2}{x_2} (-1)^{d+g-1} t^{(-g-d+2)/2} \frac{1}{1-t} \left( \frac{1-tx_2}{1-x_2} \right)^{d+g-1} \\ &= (-1)^{d+g-1} t^{(-g-d+2)/2} \frac{1 - t^{d+g-1}}{1-t} \\ &= (-1)^{d+g-1} t^{(-g-d+2)/2} \chi_{-t} \left( \mathbb{P}^{d+g-2} \right). \end{aligned} \tag{139}$$

This agrees with the description (73).

Although we will not provide a general proof of the equivalence between (138) and (75) in all chambers, it is possible to check for many values of $d$ and $g$ that these two expressions agree in all chambers for $d/2 < \widetilde{\tau} < d$[10].

---

[10]Notice in particular that the index computation (138) is similar to the derivation of [22], in that it starts from the $d - 1 < \widetilde{\tau} < d$ and corrects the results after every integer is crossed.

### 6.4.2 $\chi(\mathfrak{M}_\tau, \mathcal{K}^n)$

Let us now consider the twisted index of the theory (110) with $p \in \mathbb{Z} + \frac{1}{2}$. This can be written as

$$I(\tau, d) = \frac{1}{2} \sum_{\substack{(\mathfrak{m}_1, \mathfrak{m}_2) \in \mathbb{Z}^2 \\ \mathfrak{m}_1 + \mathfrak{m}_2 = d}} \oint_{\text{JK}} \prod_{i=1,2} \frac{dx_i}{x_i} \, Z_{CS} Z_{\text{1-loop}}(x_1, x_2, \mathfrak{m}_1, \mathfrak{m}_2) H(x_1, x_2)^g \; + \; I_{\text{boundary}}(\tau), \quad (140)$$

where the contribution from the CS level and the one-loop is

$$\begin{aligned}
Z_{CS} Z_{\text{1-loop}} = {}&(-1)^{\mathfrak{m}_1 - \mathfrak{m}_2} (x_1 x_2)^{p(g-1-d/2)} (x_1/x_2)^{3p(\mathfrak{m}_1 - \mathfrak{m}_2)/2} (x_1 - x_2)^{1-g} (x_2 - x_1)^{1-g} \\
&(x_1 x_2)^{g-1} \left( \frac{x_1^{1/2}}{1 - x_1} \right)^{\mathfrak{m}_1 + 1 - g} \left( \frac{x_2^{1/2}}{1 - x_2} \right)^{\mathfrak{m}_2 + 1 - g},
\end{aligned} \quad (141)$$

and the one-form gaugino contribution is

$$H(x_1, x_2)^g = \left[ -6p + \frac{1 + x_1}{2(1 - x_1)} + \frac{1 + x_2}{2(1 - x_2)} \right]^g. \quad (142)$$

As before, let us redefine $(\mathfrak{m}_1, \mathfrak{m}_2) \to (\mathfrak{m}, d - \mathfrak{m})$ and sum over $\mathfrak{m} \in \mathbb{Z}$ for given $d$. For each $\mathfrak{m}$, the index gets a contribution only from the boundary term, since the only rank-2 singularity of the integrand at finite $u$ is at the boundary of the Weyl chamber $x_1 = x_2 = 1$.

As discussed in section 6.3, the moduli space of the gauge theory consists of various branches depending on the value of $\tilde{\tau}$. We focus on the region $d/2 < \tilde{\tau} < d$, where the moduli space contains branch (i), which can be identified with the moduli space of the rank 2 stable pairs $\mathfrak{M}_\tau(\Lambda)$. In this region, the twisted index receives contribution from the contour integrals around the singularities at (132). In the presence of the effective CS levels (114), contribution from the first singularity in (132) is determined by the $\hat{D}_2$ integral (133) on the asymptotic components of the $\sigma_2$ integral, where $F(s, \hat{D}_2, \mathfrak{m}, \tau)$ is now given by [11]

$$F(s, \hat{D}_2, \mathfrak{m}, \tau) = \frac{\beta \text{vol}(\Sigma)}{2s^2 e^2} \hat{D}_2^2 - \frac{i\beta}{s^2} \left[ -\frac{2\pi(d - \mathfrak{m})}{e^2} + \left( \frac{(k_\pm^{\text{eff}} s^2 \sigma)_2}{2\pi} + \tau \right) \text{vol}(\Sigma) \right] \hat{D}_2. \quad (143)$$

Consider the special case $p = -\frac{1}{2}$, where

$$(k_\pm^{\text{eff}} \sigma)_2 = \begin{bmatrix} 0, & \sigma_2 > 0 \\ -\sigma_2, & \sigma_2 < 0 \end{bmatrix}. \quad (144)$$

Taking the limit $s \to 0$ sufficiently fast so that $s^2 \sigma \to \infty$, we find that the $\hat{D}_2$-integral on the contour at $\sigma_2 \to -\infty$ identically vanishes. We have

$$I(\tau, d) = \sum_{\mathfrak{m}=0}^{\infty} \Theta(d - \mathfrak{m} - \tilde{\tau}) I_{1;0}^{\mathfrak{m}} = \sum_{\mathfrak{m}=0}^{d - [\tilde{\tau}] - 1} I_{1;0}^{\mathfrak{m}}, \quad (145)$$

where we defined

$$I_{a;b}^{\mathfrak{m}} = \operatorname*{res}_{x_2 = b} \operatorname*{res}_{x_1 = a} \frac{dx_2}{x_2} \frac{dx_1}{x_1} \, Z_{CS} Z_{\text{1-loop}}(x, \mathfrak{m}) \, H(x)^g. \quad (146)$$

---

[11] The notation $(k_{\text{eff}}(\sigma)\sigma)_2$ denotes for the second component of the diagonal matrix (120).

Let us first focus on the chamber $d/2 < \widetilde{\tau} < d$, where the moduli space consists of two branches (i) and (ii). Using the residue theorem on the $x_2$-plane with $x_1$ fixed, this expression can be recast into

$$I(\tau, d) = - \sum_{\mathfrak{m}=0}^{d-\lfloor \widetilde{\tau} \rfloor - 1} \left( I_{1;1}^{\mathfrak{m}} + I_{1;\infty}^{\mathfrak{m}} \right). \tag{147}$$

From the discussion in section 6.3, we expect that the two terms on the RHS are contributions from branch (i) and branch (ii) respectively. This agrees with the fact that $\sigma_1 = \sigma_2 = 0$ on the branch (i), while in branch (ii), the solutions exist only at $\sigma_1 = 0$, $\sigma_2 \to -\infty$. These branches are non-empty in the sectors $0 \le \mathfrak{m} \le d - \lfloor \widetilde{\tau} \rfloor - 1$.

By an explicit computation, one can check that the contribution from branch (ii) is identically zero in this chamber. On the other hand, the contribution from branch (i) is expected to reproduce the holomorphic Euler characteristic of the moduli space of stable pairs

$$I(\tau, d) = \chi(\mathfrak{M}_\tau(\Lambda), \mathcal{O}). \tag{148}$$

One can check that

$$I_{1;1}^0 = 1 \quad \text{and} \quad I_{1;1}^{\mathfrak{m}} = 0, \qquad \text{for } 0 < \mathfrak{m} < d/2. \tag{149}$$

This agrees with the result obtained by taking the limit $t \to 0$ of the expression (138), which implies

$$\chi(\mathfrak{M}_\tau(\Lambda), \mathcal{O}) = 1. \tag{150}$$

Note that the index does not undergo a wall-crossing in the chambers $d/2 < \widetilde{\tau} < d$, as expected from the analysis in section 6.3. If we vary $\widetilde{\tau}$ below the critical point, $\widetilde{\tau} < d/2$, the sectors $d/2 < \mathfrak{m} < d - \lceil \widetilde{\tau} \rceil$ start to contribute and the residue integrals in these sectors are in general non-zero. This implies that the index changes discontinuously as $\widetilde{\tau}$ crosses the wall at $\widetilde{\tau} = d/2$.

Let us comment on the theories with $|p| > 1/2$. Performing the $\hat{D}$ integrals, one can show that the index can be written as a sum of two contributions

$$I(\tau, d) = I^1(\tau, d) + I^2(\tau, d), \tag{151}$$

where $I^1(\tau, d)$ is the contribution from the contours around $\sigma_1 = 0$ and $\sigma_2 \to \pm\infty$. The relevant component of the effective level on these contours is

$$(k_\pm^{\text{eff}} \sigma)_2 = \left( p \pm \frac{1}{2} \right) \sigma_2. \tag{152}$$

The $\hat{D}_2$ integral then gives,

$$I^1(\tau, d) = \sum_{\mathfrak{m}=0}^{\infty} \Theta(-p) \, I_{1;0}^{\mathfrak{m}} + \Theta(p) \, I_{1;\infty}^{\mathfrak{m}}. \tag{153}$$

Note that this expression is completely independent of $\widetilde{\tau}$, which is expected from the discussion in section 6.3. Depending on the sign of $p$, we can rewrite the integral as

$$I^1(\tau, d)|_{p<-1/2} = - \sum_{\mathfrak{m}=0}^{d-\lfloor \widetilde{\tau} \rfloor - 1} \left( I_{1;1}^{\mathfrak{m}} + I_{1;\infty}^{\mathfrak{m}} \right) + \sum_{\mathfrak{m}=d-\lfloor \widetilde{\tau} \rfloor}^{\infty} I_{1;0}^{\mathfrak{m}}, \tag{154}$$

and

$$I^1(\tau, d)|_{p>1/2} = - \sum_{\mathfrak{m}=0}^{d-\lfloor \widetilde{\tau} \rfloor - 1} \left( I_{1;1}^{\mathfrak{m}} + I_{1;0}^{\mathfrak{m}} \right) + \sum_{\mathfrak{m}=d-\lfloor \widetilde{\tau} \rfloor}^{\infty} I_{1;\infty}^{\mathfrak{m}}. \tag{155}$$

We again claim that the term $I^{\mathfrak{m}}_{1;1}$ is contribution from branch (i), and the term $I^{\mathfrak{m}}_{1;\infty}$ and $I^{\mathfrak{m}}_{1;0}$ are contributions from branch (ii). This decomposition is motivated by the fact that in the sectors $0 \le \mathfrak{m} \le d - \lfloor \widetilde{\tau} \rfloor - 1$, the solutions for $\sigma_2$ is in the region $p\sigma_2 \to \infty$, while in the sectors $\mathfrak{m} \ge d - \lfloor \widetilde{\tau} \rfloor$, the solutions for $\sigma_2$ is in the region $p\sigma_2 \to -\infty$.

The term $I^2(\tau, d)$ in (151) is the contribution from the contour which encircles the poles at $\sigma_1 \to \pm\infty$ and $\sigma_2 \to \pm\infty$. Note that for $|p| > \frac{1}{2}$, this term can be potentially non-zero, unlike the previous examples. It is natural to identify $I^2(\tau, d)$ as the contribution from branch (iii). This term is also completely independent of $\widetilde{\tau}$, which agrees with the absence of the type I wall in the entire $\widetilde{\tau}$-space, as argued in section 6.3.

One can explicitly compute the contribution from branch (i) in the chamber $d-1 < \widetilde{\tau} < d$, which can be identified with the projective space $\mathfrak{M}_0(\Lambda)$. In this case, only the $\mathfrak{m} = 0$ term contributes to the index. The pole at $x_1 = 1$ is a simple pole and one can explicitly compute the integral:

$$I^0_{1;1} = -(-1)^{d+g-1} \operatorname*{res}_{x_2=1} \frac{dx_2}{x_2} \frac{x_2^{(p+\frac{1}{2})(d+g-1)}}{(1-x_2)^{d+g-1}}. \tag{156}$$

Performing the $x_2$ integral, we find

$$I^0_{1;1} = (-1)^{d+g-1} \binom{(n-1)(1-d-g)-1}{n(1-d-g)}, \tag{157}$$

with $n = p + \frac{1}{2}$. This formula precisely agrees with the holomorphic Euler characteristic of the moduli space $\mathfrak{M}_0(\Lambda)$

$$I^0_{1;1} = (-1)^{d+g-1} \chi(\mathbb{P}^{d+g-2}, \mathcal{K}^n). \tag{158}$$

It would be very interesting to provide an explicit geometric interpretation of the contributions from branches (ii) and (iii), based on the analysis in section 6.3. We leave this for a future work.

### 6.4.3 Higher rank generalization

The Hirzebruch genus of the moduli space of rank $N$ pairs for $N > 2$ can in principle be computed using the $U(N)$ gauge theory description (78) (or (110)). We have

$$I(\tau, d, N) = \frac{1}{N!} \sum_{\substack{\underline{\mathfrak{m}} \in \mathbb{Z}^N \\ \operatorname{tr}(\underline{\mathfrak{m}})=d}} \prod_{i=1}^{N} \frac{dx_i}{x_i} Z_{\text{1-loop}}(\{x_i\}, \underline{\mathfrak{m}}) H(\{x_i\})^g + I_{\text{boundary}}(\tau), \tag{159}$$

where the one-loop contribution is simply

$$Z_{\text{1-loop}} = t^{\frac{n_0}{2}} (1-t^{-1})^{-g} (1-t^{-1})^{N(g-1)} \prod_{\alpha \in \Delta} \frac{(1-x^\alpha)^{\alpha(\underline{\mathfrak{m}})+1-g}}{(1-x^\alpha t^{-1})^{\alpha(\underline{\mathfrak{m}})+1-g}} \prod_{i=1}^{N} \left( \frac{x_i - t}{1-x_i} \right)^{\mathfrak{m}_i+1-g}, \tag{160}$$

with $n_0 = -g + (N^2 + N)(g-1) - d$. The expression for $H(\{x_i\})$ is given in (127). Again, $I_{\text{boundary}}$ is the contribution from the classical Coulomb branch boundary which encodes the dependence in $\tau$. The geometry of the boundary is in general complicated and we leave the detailed analysis to future work. For $N > 2$, the index (159) gets contribution from infinitely many GNO flux sectors labeled by $\underline{\mathfrak{m}}$ for finite $\tau$ and $d$, unlike the rank-2 case studied above. It would be interesting to work out the details and compare the $\chi_{-t}$ genus computed in [29] for the moduli space of rank-3 pairs and generalise the formula to the moduli space of higher rank pairs.

## Acknowledgments

It is a pleasure to thank Cyril Closset, Stefano Cremonesi, Tudor Dimofte, Daniele Dorigoni and Hans Jockers for fruitful discussions. The work of M.B. is supported by the EPSRC Early Career Fellowship EP/T004746/1 "Supersymmetric Gauge Theory and Enumerative Geometry". A.F. acknowledges support from the SNF Doc.Mobility fellowship P1SKP2 181340 "Twisted Hilbert Spaces of 3D Supersymmetric Gauge Theories". The work of H.K. is supported by ERC Consolidator Grant 682608 "Higgs bundles: Supersymmetric Gauge Theories and Geometry (HIGGSBNDL).

## A    Generalized vortex equations

Generalized vortex equations on a Riemann surface have been extensively studied, and their moduli spaces of solutions have been given an algebraic description by means of an extension of the classical Hitchin-Kobayashi correspondence [30–33]. This correspondence relates holomorphic vector bundles that satisfy a stability condition to Einstein-hermitian vector bundles. We recall that the latter are complex vector bundles endowed with a hermitian metric, whose curvature (seen as an endomorphism of the tangent bundle) is a constant times the identity operator. Similarly, the generalized vortex equations can be formulated as equations for the existence of a specific hermitian metric on a complex vector bundle, and Einstein-hermitian metrics can be interpreted as a special case of these.

The aim of this appendix is to summarize and develop the main notions concerning generalized vortex equations needed in the bulk of the article.

### A.1    Abelian vortex equations

Let us start with the simplest example. Consider an hermitian line bundle $L$ on $\Sigma$ endowed with a smooth unitary connection $A$. Let $\phi$ be a smooth section of $L$. The space of pairs $(A, \phi)$ that are solutions to the vortex equations on $\Sigma$

$$*F + e^2(|\phi|^2 - \tau) = 0, \qquad \bar{\partial}_A \phi = 0, \tag{161}$$

will be denoted by $\mathfrak{V}_d$. Here $F$ is the curvature of the connection $A$ and $\bar{\partial}_A$ is the holomorphic structure on $L$ determined by $d_A$ and the complex structure on $\Sigma$. We will also denote by $\mathcal{G}$ be the group of gauge transformations, $\mathcal{G} := \mathrm{Hom}(\Sigma, U(1))$. By definition the moduli space of vortices is

$$\mathfrak{M}_d := \mathfrak{V}_d / \mathcal{G}. \tag{162}$$

This can be understood as an infinite-dimensional Kähler quotient. In fact, if we consider the space of pairs $(A, \phi)$ as a Kähler manifold with flat metric

$$g = \frac{1}{4\pi} \int_\Sigma \left( \frac{1}{e^2} \delta A \wedge *\delta A + *|\delta \phi|^2 \right) d\Sigma, \tag{163}$$

then the moment map for the action of gauge transformations on the Kähler subspace of pairs satisying $\bar{\partial}_A \phi = 0$ is

$$\frac{1}{e^2} *F + |\phi|^2. \tag{164}$$

In this paper we make use of the Hitchin-Kobayashi correspondence to express the moduli space of solutions algebraically. First, we notice that by integrating the first vortex equation

in (161), a necessary condition for the existence of solutions is

$$\tau \geq \frac{2\pi d}{e^2 \text{Vol}(\Sigma)}\,. \tag{165}$$

We assume the strict version of this inequality in what follows. It is then clear that the section $\phi$ cannot vanish everywhere on $\Sigma$, which is is the simplest instance of a stability condition.

The general strategy of the Hitchin-Kobayashi correspondence for vortices on $\Sigma$ is to replace (161) with its respective stability condition, and then to take the quotient of the solution to the remaining one by complex gauge transformations $\mathcal{G}_{\mathbb{C}} = \text{Hom}(\Sigma, \mathbb{C}^*)$. The precise statement in this case [34] is that given a pair $(A, \phi)$ such that $\phi$ is a non-vanishing holomorphic section of $L$, in each complexified gauge orbit there exists one pair satisfying (161), which is unique up to $U(1)$ gauge transformations $\mathcal{G}$. Furthemore, provided the strict version of (165) holds, any solution can be written in this way.

The relation to the classical Hitchin-Kobayashi correspondence comes from the fact that (161) can be viewed as an equation for a hermitian metric $h$, intead of a connection $A$. This is because given a complex structure $\bar{\partial}_A$ and a hermitian metric $L$ there is a unique connection $A$, the Chern connection, compatible with both structures. The proof relies on this point of view, and can be applied also to the case of more general gauge groups. Finally, we remark that this construction can be viewed as an infinite-dimensional analogue of the Kempff-Ness theorem, applied to the Kähler quotient $\mathfrak{M}_d = \mathfrak{N}_d /\!/ \mathcal{G}$.

The Hitchin-Kobayashi correspondece implies that the moduli space of solutions to the vortex equations can be parametrized by pairs $(L, \phi)$, where $L$ is a holomorphic line bundle of degree $d$ and $\phi$ is a non-vanishing holomorphic section of $L$. Denoting the degree of the line bundle by $d$, there is a map from this space to the symmetric product of the curve $\text{Sym}^d \Sigma$. In fact, this parametrizes degree $d$ divisors on $\Sigma$, and the map is given by taking the divisor of zeros of $\phi$

$$D = p_1 + \ldots + p_d\,. \tag{166}$$

From a physical perspective, the points $p_1, \ldots, p_d$ correspond to the positions of the vortex centres. It turns out that the hermitian line bundle can be recovered as by means of the map

$$\begin{aligned} j : \text{Sym}^d \Sigma &\to \text{Pic}^d(\Sigma) \cong J_\Sigma \\ : \{D\} &\mapsto \mathcal{O}_\Sigma(D)\,. \end{aligned} \tag{167}$$

The connection $A$ is then defined uniquely. Thus, we have an isomorphism

$$\mathfrak{M}_d \cong \text{Sym}^d \Sigma\,. \tag{168}$$

We notice that the map $j$ has remarkable properties. Whenever $d \geq 2g - 1$ it is a holomorphic fibration with the projective space of global sections $\mathbb{P}H^0(C, L) \cong \mathbb{CP}^{d-g}$ as fibres.

## A.2 $U(N)$ vortices with fundamental matter

We now extend our discussion about the Hitchin-Kobayashi correspondnece to $U(N)$ vortices with fundamental matter. Let $E$ be a holomorphic vector bundle with structure group $U(N)$, endowed with a d-bar operator $\bar{\partial}_E$. Let $\phi \in H^0(\Sigma, E)$, that is

$$\bar{\partial}_E \phi = 0\,. \tag{169}$$

As explained in the section about Abelian vortices, it is convenient to view the vortex equation as an equation for the metric $h$. For fundamental matter, we have the $\mathfrak{u}(N)^*$-valued equation

$$*F + e^2 \left( \phi \cdot \phi^\dagger - \tau \right) = 0\,. \tag{170}$$

In analogy to the $U(1)$ case, we would like to derive a stability condition from this equation. Our presentation is based on [16]. We again integrate over $\Sigma$ and we get

$$\mu(E) \leq \frac{e^2 \tau \mathrm{vol}(\Sigma)}{2\pi}, \tag{171}$$

where for any bundle $E$ $\mu(E) := \frac{\deg(E)}{\mathrm{rank}(E)}$ is the *slope*. This is a first necessary condition, which we are now going to refine. Suppose there is a given holomorhic subbundle $E' \subset E$. As smooth complex vector bundles, we have

$$E =_{\mathrm{smoothly}} E' \oplus \left(E/E'\right), \tag{172}$$

but this might not be true holomorphically. In fact, let $(\vec{e}_1, \vec{e}_2)$ be a holomorphic unitary frame so that $\vec{e}_1$ is a basis for $E'$. Let $D_A$ be the metric connection, with respect to the metric $h$. We have

$$D_A e_a = A_{ab} e_b \tag{173}$$

where

$$A = \begin{pmatrix} A' & B \\ -B^\dagger & A^\perp \end{pmatrix}. \tag{174}$$

Here $A'$ is the metric connection that arises from the restriction of $h$ and $\bar{\partial}_E$ to $E'$ and $A^\perp$ gives a connection on the complement of $E'$. $B$ is a $(1,0)$-form which is interepreted as the second fundamental form of the embedding $E' \hookrightarrow E$ (that is, it computes the extrinsic curvature of $E'$ in $E$) and $A^\dagger$ is its conjugate transpose. [12] In obvious notation, we can compute

$$F = dA - A \wedge A = \begin{pmatrix} F' + B \wedge B^\dagger & * \\ * & F^\perp + B^\dagger \wedge B \end{pmatrix}. \tag{175}$$

Importantly, a quick computation in local coordinates shows that

$$\begin{aligned} \int_\Sigma \mathrm{Tr}\left(*B \wedge B^\dagger\right) d\Sigma &\geq 0, \\ \int_\Sigma \mathrm{Tr}\left(*B^\dagger \wedge B\right) d\Sigma &= -\int_\Sigma \mathrm{Tr}\left(*B \wedge B^\dagger\right) d\Sigma \leq 0, \end{aligned} \tag{176}$$

where the only important thing to keep in mind is that $B$ is of type $(1,0)$. Now, we can also write (170) in local coordinates as

$$\begin{pmatrix} *F' + *B \wedge B^\dagger & * \\ * & F^\perp + *B^\dagger \wedge B \end{pmatrix} + e^2 \begin{pmatrix} \phi' \phi'^\dagger - \tau & * \\ * & \phi^\perp \phi^{\perp\dagger} - \tau \end{pmatrix} = 0. \tag{177}$$

Taking the trace of the upper left component and integrating over the curve, we get that

$$\mu(E') \leq \frac{e^2 \tau \mathrm{vol}(\Sigma)}{2\pi}, \tag{178}$$

with equality if and only if

$$\int_\Sigma \mathrm{Tr}\left(*B \wedge B^\dagger\right) d\Sigma = 0. \tag{179}$$

By definition, if the above equation holds, then

$$E =_{\mathrm{hol}} E' \oplus \left(E/E'\right) \tag{180}$$

---

[12]$B = (1-\pi)D_A \pi$ where $\pi$ is the projection onto $L$. Since $L$ is a holmorphic subbundle, $B$ must be a $(1,0)$-form.

holomorphically. Now suppose that $\phi \in H^0(\Sigma, E')$. Then, taking the trace of the lower-right component of (177), we similarly get

$$\mu\left(E/E'\right) \geq \frac{e^2 \tau \mathrm{vol}(\Sigma)}{2\pi}, \tag{181}$$

with equality if and only if (179) holds, $E/L$ is holomorphic and (180) holds holomorphically.

We can now summarize the above findings as follows. Let

$$
\begin{aligned}
\mu_M &:= \sup\{\mu(E),\ \mu(L) \mid L \text{ holomorhic subbundle of } E\}, \\
\mu_m &:= \inf\{\mu(E/L) \mid L \text{ holomorhic subbundle of } E, \phi \in H^0(\Sigma, L)\}.
\end{aligned}
\tag{182}
$$

Further, define the following notion of stability for pairs $(E, \phi)$

**Definition 1** *A pair $(E, \phi)$ is stable if and only if*

$$\mu_M < \frac{e^2 \tau \mathrm{vol}(\Sigma)}{2\pi} < \mu_m.$$

Then we have the following

**Lemma 1** *If there is a metric $h$ satsifying the equations (170), then either the pair $(E, \phi)$ is stable or $E =_{hol} E' \oplus \left(E/E'\right)$ with $\phi \in H^0\left(\Sigma, E'\right)$. In the latter case, the pair $(E', \phi)$ satisfies the inequality*

$$\mu(E') < \frac{e^2 \tau \mathrm{vol}(\Sigma)}{2\pi},$$

*and the holomorphic bundle $E/E'$ satisfies*

$$\mu\left(E/E'\right) = \frac{e^2 \tau \mathrm{vol}(\Sigma)}{2\pi}.$$

In [16] the converse is also proven.

Finally, we consider the $|\tau| \to \infty$ limit. In this limit, the stability condition simplifies drastically. First of all, notice that in this limit the lower bound is obviously satisfied. As for the upper bound, it is easy to see that it immediately implies that $\phi$ cannot be contained in any subbundle of $E$. But this means that generically $\phi$ has maxiaml rank. This discussion can be generalized to matter fields in both the fundamental and anti-fundamental representation at no cost. The result in the large $\tau$ limit remains the same. For other representations, more sophisticated techniques are needed [32, 35].

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
