# Peer review of "The 3d Twisted Index and Wall-Crossing"

_SciPost Physics, doi:SciPost Phys. 12, 186 (2022)_

## Round 1 · Referee Report · Anonymous (Referee 1) · 2022-3-22

Strengths

  1. Exhibits an interesting wall-crossing phenomenon w.r.t. to a 1d FI parameter upon reducing 3d N=2 gauge theories on S^1x\Sigma to quantum mechanics on S^1
  2. Explains two perspectives of this phenomenon by looking at a Higgs and Coulomb branch localization of the supersymmetric quantum mechanical setup.
  3. Illustrates the obtained phenomenon with explicit examples.

Weaknesses

  1. The summation/regularization in comparison with the previous result upon summing over all sectors remains a bit mysterious.
  2. The quantum mechanics perspective does not address a subtlety for non-Abelian gauge groups and for genus(Sigma)>0. In this case the vector multiplets gives rise to additional poles.

Report

The submitted manuscript is an impressive work exhibiting an interesting wall-crossing phenomena upon reducing 3d N=2 gauge theories to quantum mechanics. The wall-crossing phenomena arises from a 1d FI parameter in the quantum mechanical description. The phenomenon is explained abstractly both from a Coulomb branch and Higgs branch localization perspective. The Higgs branch localization offers an interesting geometric explanation in terms of (virtual) Euler characteristics of the defined sectors of the moduli space. The general features are impressively illustrated with examples (including a demanding analysis of a non-Abelian example).

---

## Round 1 · Referee Report · Anonymous (Referee 2) · 2022-3-25

Strengths

1 Comprehensive analysis of a new phenomenon for 3D N=2 theories.

2 Compelling agreement between Higgs and Coulomb branch approaches in the abelian case.

3 High standard of pedagogy.

Weaknesses

1 Lack of contextual discussion of results and motivation.

Report

In this interesting paper the authors investigate a (to me) novel form of wall -crossing for the twisted index of N=2 SUSY theories in three dimension. These indices have dual representations in terms of the so-called Higgs and Coulomb branch localisation and the authors perform a thorough investigation of wall crossing both approaches. As in other related situations wall crossing is manifested in the Coulomb branch approach by a change in the JK contour prescription, while on the Higgs branch it reflects the change in stability conditions in the algebraic description of the associated vortex moduli spaces. In the abelian case, where the vortex moduli spaces are essentially symmetric products the authors push both sides to completion and find nice agreement between the two pictures. Some preliminary results for the non-abelian generalisation are also presented.

While this is very nice, I did not feel the authors did much to motivate this research or to relate it to other related developments. In particular,
there are strong parallels with wall-crossing for Donaldson Thomas invariants seen where the generating function of the DT invariants has a similar interpretation in terms of an index of the D0 brane quantum mechanics. In this case wall crossing is essentially determined by the powerful quantum group based approach of Kontsevich and Soibelman and it’s interesting to ask whether something similar is present here. This seems like an obvious reference point for the present work. Another is the interesting work (some of it by at least one of the present authors) on the relation of these indices and their blocks (in the N=4 case) to the quantised algebras of the Higgs and Coulomb branches.
It would have been nice to see more discussion of these points or perhaps some other motivation that the authors had in mind.

The paper meets the expectations for a Scipost publication. Criterion 3 which refers to establishing a new direction is the most relevant. The list of general criteria is also satisfied.

---

## Round 2 · Referee Report · Anonymous (Referee 2) · 2022-5-2

Report

This paper is now suitable for publication

---

## Round 2 · Referee Report · Anonymous (Referee 1) · 2022-5-2

Report

I support the resubmitted manuscript with the described modifications for publication.

---

## Round 2 · Author Response

With this resubmission we address the points raised by the referees. In particular, we add some context and emphasize the relation of our prescription to the previous JK prescription. We acknowledge the fact that subtleties remain for non-Abelian gauge groups at higher genus due to the presence of additional poles; this was already pointed out in section 3.1, page 10 and in footnote 9, page 34.

---

## Round 2 · List of Changes

We added some context and motivation in the introduction and emphasized the relation to the previous prescription in subsection 3.4.

---

## Editorial Decision

published